



# Modelling framework for asynchronous land-atmosphere coupling using NASA GISS ModelE and LPJ-LMfire: Design, Application and Evaluation for the 2.5ka period

Ram Singh[1,2], Alexander Koch[4], Allegra N LeGrande[2,1], Kostas Tsigaridis[1,2], Riovie D Ramos[5], Francis Ludlow[6], Igor Aleinov[1,2], Reto Ruedy[2,3], Jed O Kaplan[7]

[1] Center for Climate Systems Research, Columbia University, New York, USA

[2] NASA Goddard Institute for Space Studies, New York, NY-10025, USA

[3] SciSpace LLC, New York, NY, USA

[4] Department of Earth Sciences, The University of Hong Kong, Hong Kong SAR, China

[5] Earth Observatory of Singapore, Singapore

[6] Department of History, School of Histories and Humanities, Trinity College, Dublin 2, Ireland

[7] Department of Earth, Energy, and Environment, University of Calgary, Calgary AB, Canada

*Correspondence to*: Ram Singh ( ram.bhari85@gmail.com)





**Abstract**
While paleoclimate simulations have been a priority for Earth system modelers over the past
three decades, little attention has been paid to the period between the mid-Holocene and the Last
Millennium, although this is an important period for the emergence of complex societies. Here,
we consider the climate of 2500 BP (550 BCE), a period when compared to late preindustrial
time, greenhouse gas concentrations were slightly lower, and orbital forcing led to a stronger
seasonal cycle in high latitude insolation. To capture the influence of land cover on climate, we
asynchronously coupled the NASA GISS ModelE Earth system model with the LPJ-LMfire
dynamic global vegetation model. We simulated global climate and assessed our results in the
context of independent paleoclimate reconstructions. We also explored a set of combinations of
model performance parameters (bias and variability) and demonstrated their importance for the
asynchronous coupling framework. The coupled model system shows substantial vegetation
albedo feedback to climate. In the absence of a bias correction, while driving LPJ-LMfire in the
coupling process, ModelE drifts towards colder conditions in the high latitudes of the Northern
Hemisphere in response to land cover simulated by LPJ-LMfire. A regional precipitation
response is also prominent in the various combinations of the coupled model system, with a
substantial intensification of the Summer Indian Monsoon and a drying pattern over Europe.
Evaluation of the simulated climate against reconstructions of temperature from multiple proxies
and the isotopic composition of precipitation ($\delta^{18}O_p$) from speleothems demonstrated the skill of
ModelE in simulating past climate. A regional analysis of the simulated vegetation-climate
response further confirmed the validity of this approach. The coupled model system is sensitive
to the representation of shrubs and this land cover type requires particular attention as a
potentially important driver of climate in regions where shrubs are abundant. Our results further
demonstrate the importance of bias correction in coupled paleoclimate simulations.



## 1. Introduction

Earth system models (ESMs) are widely applied in paleoclimate experiments as an "out of
sample" exercise to evaluate the overall quality of the model, and to better understand climate
system responses to external forcings. In many paleoclimate modeling studies, it has been
demonstrated that inclusion of biogeophysical and biogeochemical feedbacks between land and
atmosphere feedbacks are essential to simulate the magnitude and spatial pattern of climate
change that is consistent with independent reconstructions (Betts, 2000; Claussen, 1997; Cox et
al., 2000; Doherty et al., 2000; Strandberg et al., 2014). The importance of land-atmosphere
feedbacks for past climate has shown particularly to be true in the context of the mid-Holocene
and last glacial inception periods (Braconnot et al., 2012; Collins et al., 2017; Harrison et al.,
2015; Jahn et al., 2005; Kubatzki and Claussen, 1998; Sha et al., 2019; Shanahan et al., 2015;
Tierney et al., 2017). For example, for the African Humid Period of the mid-Holocene, numerous
studies demonstrated that greenhouse gases ($CO_2$, $N_2O$, $CH_4$) and orbital forcing are alone not
sufficient for models to simulate climate that is consistent with independent paleoclimate
reconstructions. The inclusion of land-atmosphere feedbacks via interactive dynamic vegetation
modeling or prescribed vegetation distributions helps improves model-proxy discrepancies
(Chandan and Peltier, 2020; Charney, 1975; Dallmeyer et al., 2021; Pausata et al., 2016;
Rachmayani et al., 2015; Singh et al., 2023; Thompson et al., 2021; Tiwari et al., 2023;
Velasquez et al., 2021). For this reason, more recent protocols (PMIP4; Otto-Bliesner et al.,
2017) for simulations of the mid-Holocene specify that the land cover boundary condition should
include shrub vegetation in northern Africa with greater extent than the present (the so-called
"Green Sahara"), as well as an expansion of trees and shrubs at high northern latitudes.

Instead of prescribing land cover boundary conditions in an earth system model, it may be
desirable to employ a coupled model where that allows interaction between climate and
vegetation. While several modern earth system models include a dynamic representation of land
cover, in climate models (regional and global) that lack a coupled dynamic vegetation
component a well-established technique to capture land-atmosphere feedbacks is to use
asynchronous coupling. In this type of coupling, climate model output is used to drive an offline
vegetation model that then returns a land cover boundary condition to the climate model.





To quantify the feedback between land and atmosphere and improve the fidelity of the
paleoclimate simulation, asynchronous coupling typically involves running a climate model
simulation for a period of a few decades, after which the mean climate state is passed to a
vegetation model that in-turn produces a land cover boundary condition for the climate model.
This process is repeated until climate reaches equilibrium, defined as insignificant changes in
key outputs, e.g., 2m temperature, from one cycle to the next.

Texier et al. (1997) used the iterative asynchronous coupling between the LMD Atmospheric
General Circulation Model (AGCM) and the BIOME1 vegetation model to produce an improved
climate for the mid-Holocene (6ka) period and found that inclusion of land-atmosphere
feedbacks led to simulations of temperatures at high latitudes and precipitation over West Africa
that were more consistent with independent paleoclimate reconstructions compared to
atmosphere-only simulations. de Noblet et al. (1996) used a similar coupling to highlight the role
of biogeophysical feedback in glacial initiation around 115ka ago. Asynchronous coupling has
also been used with regional climate models (RCMs). Kjellstrom et al. (2008) and Velasquez et
al. (2021) both used asynchronous coupling between an RCM and land cover model to simulate
the climate of Europe at the Last Glacial Maximum. Both studies demonstrated the importance
of land cover in improving the agreement with reconstructions and paleoenvironmental proxies.

This study has two objectives. First, we present a generalized design for asynchronously
coupling the NASA GISS ModelE2.1 climate model (Kelley et al., 2020) with the LPJ-LMfire
DGVM (Pfeiffer et al., 2013) to simulate climate including biogeophysical land-atmosphere
feedbacks. Second, we demonstrate the utility of this asynchronous coupling framework for a
paleoclimate period that has not been the traditional focus of paleoclimate modeling (2.5 ka) and
evaluate the model results against independent paleoclimate reconstructions for that period.

2.5 ka represents a time that is nearest to the present day among the different periods selected
under the coordinated effort of the Paleoclimate Model Intercomparison Project (PMIP4). It is
interesting because it represents an important period for the emergence of complex societies
across Eurasia (Iron Age, Classical Antiquity, early Imperial China) and elsewhere. During this
era, favorable climate conditions around the Mediterranean might have influenced the emergence



of the golden age of Greece, the Roman classical period, and other empires of the Southern
Europe, North Africa, and southwest Asia (Lamb, 1982; Reale and Dirmeyer, 2000). On the
other hand, adverse climate conditions due to volcanic eruptions and a series of arid phases
during this period may have had a negative impact on Egyptian civilization around the Nile and
Mesopotamian civilization around the Euphrates and Tigris rivers. 2.5ka is thus a key period for
the study of human-environment interactions and the history of climate and society, where we
may assess societal vulnerability to climate change (Ludlow and Manning, 2021; Manning et al.,
2017; Mikhail, 2015; Petit-Maire and Guo, 1998; Singh et al., 2023).

We evaluate the climate of 2.5 ka simulated with the ModelE-LPJ asynchronous coupling
framework against multi-proxy temperature reconstructions (Kaufman et al., 2020) and
additionally utilize the model's capabilities to simulate the isotopic composition of water in
precipitation ($\delta^{18}Op$) to compare with the Speleothem Isotope Synthesis and Analysis (SISAL)
version 2 database (Comas-Bru et al., 2020).

## 2. Models and Methodology

**2.1.1 NASA GISS ModelE2.1**: NASA GISS ModelE2.1 (Kelley et al., 2020), is the climate model
of the NASA Goddard Institute for Space Studies (GISS) currently used in Climate Model
Intercomparison Project (CMIP) phase 6 (Eyring et al., 2016). We used the NINT (Non-
Interactive; physics version 1 in CMIP6) GISS ModelE2.1 version where aerosols and ozone are
precomputed from the prognostic, but much more computationally demanding, chemistry and
aerosols version of the model OMA (One Moment Aerosols; physics version 3 in CMIP6; (Bauer
et al., 2020)). In our simulations, the GISS ModelE2.1 atmosphere has a horizontal resolution of
2°x2.5° (latitude/longitude) with 40 vertical layers, and the top of the atmosphere at 0.1 hPa. The
ModelE2.1 atmosphere has a smooth transition from sigma layers to constant pressure layers
centered at 100hPa. The atmosphere is coupled to the GISS Ocean v1 model, which runs at a
resolution of 1°x1.25° (latitude/longitude) with 40 depth layers to the ocean bottom. While the
biogeophysical properties of land cover are simulated with the Ent Terrestrial Biosphere Model
(Ent TBM; Kiang 2012; (Kim et al., 2015)), as part of ModelE2.1 (Ito et al., 2020), Ent relies on
a prescribed vegetation map and as such does not simulate changes in land cover over time. To
capture the influence of climate change on land cover and biogeophysical feedbacks between land





and atmosphere, asynchronous coupling with LPJ-LMfire (or any other DGVM) is currently
required.

**2.1.2 LPJ-LMfire:** We used the LPJ-LMfire DGVM (v1.4.0) to simulate the land cover
boundary conditions in our experiments. LPJ-LMfire (Kaplan et al., 2022; Pfeiffer et al., 2013) is
an evolution of LPJ (Sitch et al., 2003) and is a process-based, large-scale representation of plant
growth and decay, vegetation demographics and ecological disturbance, and water and carbon
exchanges between the land and the atmosphere. For this study, we simulated land cover
boundary conditions at a horizontal resolution 0.5°x0.5°. LPJ-LMfire is driven by monthly fields
of climate (temperature, precipitation, cloud cover, wind, and lightning), static maps of
topography and soil texture, and an annual global value of atmospheric $CO_2$ concentration. LPJ-
LMfire simulates land cover in the form of fractional coverages of nine plant functional types
(PFTs), including tropical, temperate, and boreal trees, and tropical and extratropical herbaceous
vegetation (Table 1). $CO_2$, soil texture and topography data used to drive LPJ-LMfire are
described in Pfeiffer et al. (2013, Table 3). For 2.5ka simulations, we set atmospheric CO2
concentrations to 271.4 ppm (Krumhardt and Kaplan, 2012). The sum of PFT fractional cover
per grid box does not need to equal unity; when it is less than one the remainder is considered
bare ground.

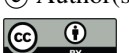




| GISS Output | LPJ -LMfire Input | Annual cycle climatology and variability (standard deviation) over the period of interest (100 Years) | LPJ-Lmfire Output Vegetation (PFTs) | LPJ-LMfire to GISS ModelE (Ent) Vegetation Mapping (Vegetation cover type, Leaf area index and vegetation heights) | GISS ModelE (Ent) Vegetation (PFTs) |
|---|---|---|---|---|---|
| Surface Air Temperature | Surface Air Temperature | | Tropical Broadleaf Evergreen | | Evergreen Broadleaf Late Succession |
| Precipitation | Precipitation | | Tropical Broadleaf Raingreen | | Evergreen Needleleaf Late Succession |
| | Number of wet days | | | | |
| Diurnal Surf. Air Temp Range | Diurnal Surf. Air Temp Range | | Temperate Needleleaf Evergreen | | Cold Deciduous Broadleaf Late Succession |
| Surface Wind Speed | Surface Wind Speed | | Temperate Broadleaf Evergreen | | Drought Deciduous Broadleaf |
| Moist Convective Air Mass Flux | Lightning Density | | Temperate Broadleaf Summergreen | | Deciduous Needleleaf |
| | | | Boreal Needleleaf Evergreen | | Cold Adapted Shrub |
| | | | Boreal Summergreen | | Arid Adapted Shrub |
| | | | C3 Perennial Grass | | C3 Grass Perennial |
| | | | C4 Perennial Grass | | C4 Grass |
| | | | | | C3 Grass Annual |
| | | | | | Arctic C3 Grass |
| | | | | | Bright Bare Soil |
| | | | | | Dark Bare Soil |






**Table 1: -** Summary of climate and PFT variables exchanged between NASA GISS ModelE and LPJ-LMFire model for asynchronous coupling process. Column 1 and 2 shows lists the output and input climate variables from GISS ModelE to LPJ-LMFire models, whereas the columns 3 and 4 lists the output and input plant function types (PFTs) from LPJ-Lmfire to GISS ModelE.

## 2.2. 2.5ka Simulation setup (ModelE)

We started the 2.5ka and preindustrial (PI) control experiments following the PMIP4 and CMIP6 protocols (Eyring et al., 2016; Kageyama et al., 2018). The PI simulation uses preindustrial (year 1850) GHG concentrations and a modern continental configuration and serves as the reference experiment for designing the boundary conditions for past time slices studied in PMIP4. GHG and orbital forcings for the preindustrial (PI) control experiment correspond to levels observed in 1850 CE ($CO_2$: 284 ppm, $N_2O$: 273 ppb, $CH_4$: 808 ppb). For the 2.5 ka control experiment, orbital parameters (Berger et al., 2006) were specified for 2,500 years BP (~550 BCE), and greenhouse gas $CO_2$, $N_2O$, and $CH_4$ were set to ~279 ppm, ~266 ppb, and 610 ppb respectively (Loulergue et al., 2008; Otto-Bliesner et al., 2017; Schneider et al., 2013; Siegenthaler et al., 2005). We considered only natural emissions as sources of aerosols in the atmosphere, zeroing-out any anthropogenic contribution to aerosol and aerosol precursors. For biomass burning, in the absence of any better estimate, we assumed that the emissions provided by CEDS (Hoesly et al., 2018) for the year 1750 are all natural. Land cover consists of the fractional coverages of 13 plant functional types (PFTs) and includes vegetation height and leaf area index (LAI). For the PI and initial (0[th] order) simulations, land cover type and monthly-varying LAI were derived from satellite (MODIS) data (Gao et al., 2008; Kattge et al., 2011; Myneni et al., 2002; Tian et al., 2002a, b; Yang et al., 2006) and vegetation heights from (Simard et al., 2011). We also used the mid-Holocene (6k) vegetation under PMIP4 protocol, which is linearly interpolated to 2.5ka period and details of vegetation cover changes (Singh et al., 2023; Figure S1) and associated impacts on the northern hemisphere climate due to the inclusion of scaled PMIP4 vegetation using the interactive chemistry version of NASA GISS ModelE2.1 (MATRIX) are discussed in (Singh et al., 2023).

## 2.3 Asynchronous Coupling Framework

The asynchronous coupling between ModelE and LPJ-LMfire is summarized in Figure 1. For each iteration, ModelE simulated climate is used by LPJ-LMfire, which, returns the PFT fractional





cover, LAI, and vegetation height that are used as boundary conditions for the next ModelE
simulation.

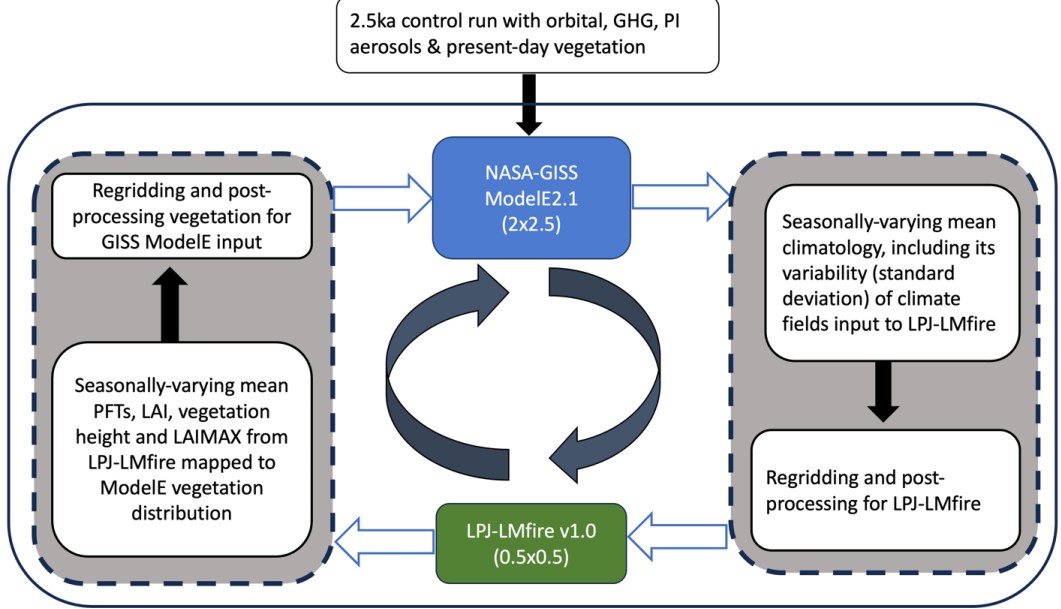



Figure 1: Flow diagram for the asynchronous coupling between GISS ModelE2.1 and LPJ-LMfire
models. For the climate fields input to LPJ-LMfire refer to (Table 1, Column 1) and LPJ-LMfire
PFTs (Table 1, Column 3)

**2.3.1 GISS ModelE2.1 simulations:** Climatological monthly mean climate (Table 1, Column 1)
for a 100-year period were extracted from a well equilibrated ModelE simulation. To assess
interannual variability with monthly resolution, we calculated the standard deviation of the decadal
mean data for each month across the 100-year equilibrium period.
**2.3.2. LPJ-LMfire simulations:** All climate variables except diurnal temperature range, wet days,
and lightning density were provided directly from the ModelE output. For derived climate
variables, the additional processing steps are described below.

Diurnal temperature range was calculated as the difference of the monthly-mean daily maximum
and minimum temperatures as simulated by ModelE. Wet days were calculated from modelled
precipitation based on an empirical relationship between present-day monthly total precipitation





and the number of wet days per month. To quantify this relationship, we performed a nonlinear
regression between monthly total precipitation and number of days with measurable precipitation
using the CRU TS 4.0 gridded climate fields (Harris et al., 2020). Using those data, we developed
a set of regression coefficients for every land gridcell that allowed us to estimate wet days for any
paleoclimate period based only on monthly total precipitation. Lightning density was estimated
based on modelled convective mass flux following Magi (2015).

Because LPJ-LMfire requires a timeseries of interannually varying climate forcing to run, we
processed the climatological monthly mean climate produced by the ModelE for use with the
vegetation model. In brief, ModelE climate was converted into anomalies by differencing the
paleoclimate simulation with ModelE simulated climate for the late 20[th] century (1951-2000). The
resulting climate anomalies were linearly interpolated to a 0.5°x0.5° grid and added to a baseline
climate based on observations over 1951-2000. The resulting climatology was expanded to a 1020-
year-long time series by adding interannual variability in the form of detrended and randomized
climate anomalies from the 20[th] Century Reanalysis (Compo et al., 2011). For further details on
this process, see (Hamilton et al., 2018). Because LPJ-LMfire is computationally inexpensive, we
ran each simulation for 1020 years. While the composition and characteristics of aboveground
vegetation comes into equilibrium with climate after a few centuries of simulation, a millennium-
long simulation brings the terrestrial carbon pools into equilibrium as well. The land cover
boundary conditions returned to the climate model represent the mean modeled vegetation cover
over the final 250 years of the LPJ-LMfire simulation.

**2.3.3. LPJ-LMfire to GISS ModelE vegetation mapping:** LPJ-LMfire simulates land cover in
the form of nine PFTs, while in GISS ModelE the vegetation component (Ent TBM) recognizes
13 PFTs. We mapped the LPJ-LMfire generated PFT cover, LAI, LAIMAX, and vegetation height
to the GISS ModelE2.1 (Ent) PFTs in order to feed it to the ModelE (Table 1, Column 3 & 4). The
main points for the LPJ-LMfire to GISS vegetation mapping are the following:

235        -   Early and late-successional PFTs were approximated from the LPJ-LMfire output using

236            the model simulated fire frequency and monthly burned area fraction. However, because

237            successional state is indistinguishable in the satellite-driven reference vegetation for the



historical period used as the boundary condition for ModelE, we combined early & late
successional PFTs in our simulations.
-   LPJ-LMfire does not have a specific PFT for shrubs (arid and cold), while Ent does. To
estimate shrub cover in LPJ-LMfire, we used LPJ-LMfire simulated tree height for the
tropical broadleaf raingreen, temperate broadleaf summergreen, and boreal summergreen
PFTs and specified that trees with height lower than a predefined threshold were considered
to be shrubs (Table S1).
-   Ent has an Arctic grass PFT while LPJ-LMfire does not. To estimate Arctic grass cover we
used the $C_3$ grass PFT in LPJ-LMfire and specified it as Arctic grass in regions where the
boreal summergreen PFT was also present. LPJ-LMfire also does not distinguish between
annual and perennial grasses, and so to map these to Ent we assumed that these were
present in equal fractions among the simulated $C_3$ grass in the LPJ-LMfire simulation.
-   The non-vegetated fraction of a grid cell is assigned to the bare soil, and the distribution of
bright and dark soil color heterogeneity is classified/redistributed based on the present-day
structure of soils over a grid cell.

Of particular importance to our coupled model simulations was that the PFTs simulated by LPJ-
LMfire do not explicitly include a shrub type. To approximately distinguish tree from shrub cover,
we generated three LPJ-to-GISS mapping schemes that differed on how shrubs are specified. A
set of possible changes in various PFT classifications are adopted based on the comparison with
GISS vegetation distribution and categorized the mapping methodologies. These mappings,
summarized in table S1, differ in the height threshold of trees to be re-categorized as cold and arid
shrubs, and the fraction of perennial grass re-categorized into perennial and arctic grasses. Also,
the monthly leaf area index (LAI) and vegetation height readjusted using the weighted mean for
remapped LPJ-LMfire vegetation PFTs.

**2.3.4. Step 4. Post-processing of vegetation files:** LPJ-LMfire model generates output at a
horizontal resolution of 0.5°x0.5°. We resampled the output vegetation information to the
2.0°x2.5° grid used by ModelE2.1, In a few cases, land cover extrapolated using a nearest-neighbor
approach was to cover all the gridcells identified as land in the ModelE standard land-sea mask.





## 3 Experimental Design


Apart from evaluating the framework for the PI control period, we designed a set of experiments
to evaluate various aspects of the simulated climate, including model bias, and variability in both
the climate vegetation models. For example, one known limitation in the current version of
ModelE is a wintertime cold bias over the Arctic in simulations covering the historical period
(Kelley et al., 2020).

Table 2 shows the combinations of the model metrics selected to explore the utility of the
asynchronous coupling framework and their impact on simulated climate. Run names are
designated using Time (1850, 2.5k), Vegetation source (PI, GS), Bias Correction (BC) and
Interannual Variability (LPJ, GISS) separated by "_". For example, '1850_PI_ctrl' and
'2.5k_PI_ctrl' denote the 1000-year-long PI and 2.5k runs with GISS PI vegetation. GS stands for
Green Sahara and PI = Pre-Industrial. An "x" denotes the absence of a particular criterion (default
state). Runs '2.5k_PI_BC_LPJ', '2.5k_PI_x_x', and '2.5k_PI_x_GISS' are three branches
extended from '2.5k_PI_ctrl' with the combinations of bias correction and interannual variability
from LPJ and GISS models. For the '2.5k_GS_x_GISS' and '2.5k_GS_BC_GISS' simulations,
we initialized the land cover boundary conditions to approximate 2.5 ka by linearly interpolating
cover fractions between the 6 ka land cover prescribed under the PMIP4 protocol (Otto-Bliesner
et al., 2017) and the PI reference dataset. Details of the 6 ka land cover boundary conditions under
for PMIP4 and associated impacts on Northern Hemisphere climate using the interactive chemistry
version of NASA GISS ModelE2.1 (MATRIX) are discussed by (Singh et al., 2023).





**Table 2: -** Summary of experiment designs followed to explore and evaluate the GISS ModelE -
LPJ-LMFire model asynchronous coupling framework. See text for an explanation on the run
naming convention.

| Run Name | Initial Vegetation Cover | Bias correction | Interannual Variability | Number of Iterations/total number of years | Remark |
|---|---|---|---|---|---|
| 1850_PI_ctrl | Used to evaluate the LPJ to GISS vegetation mapping schemes | | | | |
| 2.5k_PI_ctrl | 1000-year-long control; base run to branch out the other simulations | | | | |
| 2.5k_PI_BC_LPJ | GISS PI vegetation | YES | LPJ | 5/750 years | converged |
| 2.5k_PI_x_x | GISS PI vegetation | No | No | 2/270 years | Too cold in 3rd iteration diverging |
| 2.5k_PI_x_GISS | GISS PI vegetation | No | GISS ModelE (100years) | 4/550 years | Too cold diverging |
| 2.5k_GS_x_GISS | GISS PI vegetation + Green Sahara+ Boreal Forest | No | GISS ModelE (100years) | 5/1150 years | Too cold diverging |
| 2.5k_GS_BC_GISS | GISS PI vegetation + Green Sahara+ Boreal Forest | YES | GISS ModelE (100years) | 4/1000 years | converged |

* Convergence means the final model simulation has a similar climatology with the previous
iteration, whereas divergence means the model is drifting away from the expected states.






**3.1 Evaluation & Validation of LPJ-GISS Mapping Methodologies**

We used the standard present-day land cover boundary conditions described for ModelE2.1 (Kelley et al., 2020) for the initial $0^{th}$-order iteration of the pre-industrial and 2.5ka control climate simulations. This land cover dataset is based on satellite observations (Gao et al., 2008; Myneni et al., 2002; Tian et al., 2002a, 2002b; Yang et al., 2006) from the Moderate Resolution Imaging Spectroradiometer (MODIS), with leaf area index (LAI) from the TRY database (Kattge et al. 2011), and vegetation height (Simard et al. 2011) from the Geoscience Laser Altimeter System (GLAS). Branches of the 2.5ka run for green Sahara conditions are started using the linearly interpolated vegetations for 2.5ka from the 6ka vegetation distribution defined based on the PMIP4 protocol (Otto-Bliesner et al., 2017; Singh et al., 2023). These land cover boundary conditions are shown as the fractional coverage of 13 PFTs (including bare soils) (Figs. S1.A and S1.B). In these figures, bare dark and bare bright are merged into a single bare soil fractional cover.

The ModelE2.1 pre-industrial (PI) control run initialized with the present-day land cover boundary condition is processed through the asynchronous coupling framework to evaluate the mapping scheme for converting LPJ PFTs to GISS (Ent) PFTs. We tested three sets of LPJ-to-GISS mapping schemes as required in the asynchronous coupling framework. Differences among the mapping schemes are described in supplementary table TS1. Three parallel control runs are performed for 100 years, each initialized with the vegetation distribution that corresponds to the corresponding mapping scheme and compared to the mean climate state of the parent PI control run.



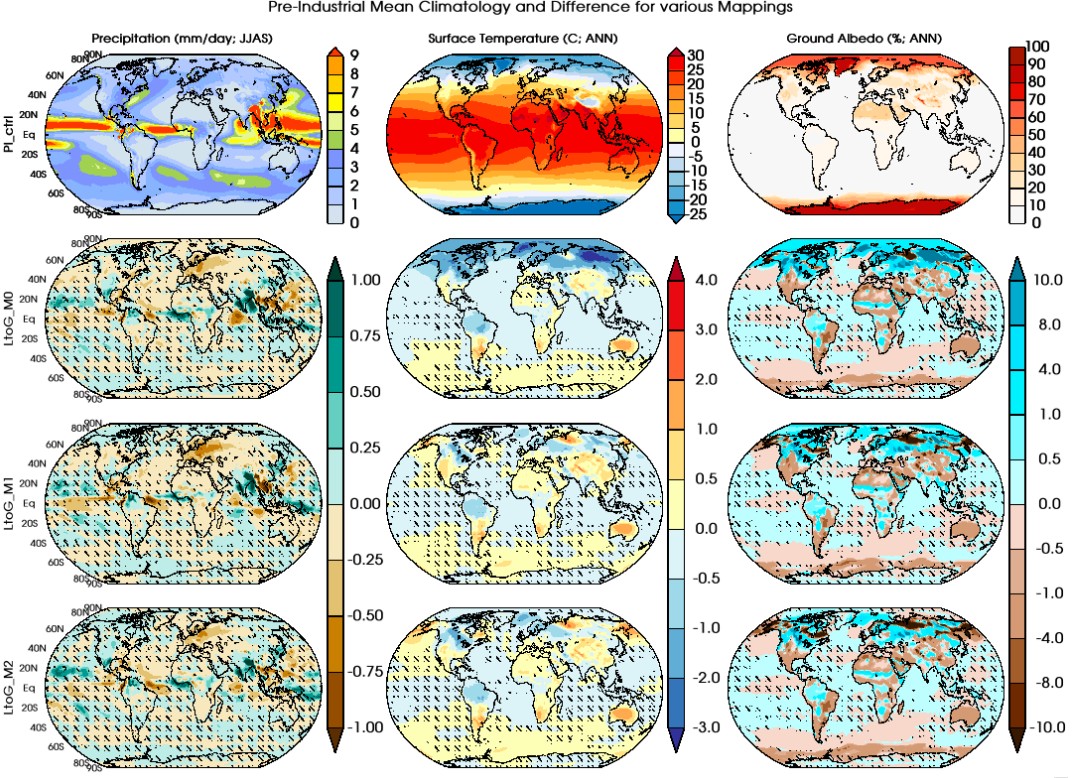

**Figure 2.** Comparison of seasonal mean climate metrics when using different vegetation mapping schemes with that of the origin PI control. Top row shows the mean climatology for precipitation (mm/day; JJAS), surface air temperature (°C; ANN) and ground albedo (%; ANN) and row 2 to 4 differences in mean climate for LtoG_M0, LtoG_M1 and LtoG_M2, respectively.

The mapping schemes LtoG_M1 and LtoG_M2 (supplementary table TS1) generate a similar spatial structure of annual surface air temperature with broadly similar regional characteristics (Fig. 2). A shift towards colder climates of 2-3 °C in mean annual temperature over the higher latitudes of the Northern hemisphere is simulated when using the mapping scheme LtoG_M0, which is not present when using the other mapping schemes (LtoG_M1 and LtoG_M2). We selected forests into shrubs to match the missing PFTs in ModelE vegetation distributions based upon the tree height (Table S1). In these mapping schemes, the fraction of boreal tree PFTs assigned to cold shrubs depends on simulated tree height, which is, in turn, influenced by surface temperature (Thomas and Rowntree, 1992; Bonan et al., 1992; 2008; Li et al., 2013). In the





mapping LtoG_M0, the fractional cover of boreal tree PFTs was reduced significantly, leading to
an increase in ground albedo (up to 10%), which led to the model drifting towards comparatively
colder climate conditions. When using the other two mapping schemes (LtoG_M1 and
LtoG_M2) the assignment of boreal tree PFTs to shrub types is limited by a higher tree height
threshold and partially because other PFTs (perennial grass) are substituted for cold shrubs.
Regional patches of increased ground albedo and surface cooling over the higher latitudes of the
Northern Hemisphere are also evident when using the LtoG_M1 and LtoG_M2 translation
schemes.

Precipitation during the Northern Hemisphere summer monsoon season (JJAS; June-July-
August-September) appears similar among the three mapping schemes, as the larger changes are
confined to the equatorial regions. A drying pattern over Europe appears in all three translation
schemes, but it is comparatively more substantial under LtoG_M0 and LtoG_M1 than LtoG_M2.

All translation schemes also lead to increased precipitation over equatorial South America.
Annual mean river runoff for the Amazon River is simulated at 305, 297, and 308 $km^3$/month for
LtoG_M0, LtoG_M1 and LtoG_M2, respectively, a slight improvement to the original
Preindustrial (PI) run runoff of 280 $km^3$/month with using the standard present-day land cover
boundary condition. Compared to observations, ModelE2.1 shows a substantial deficit in
Amazon River runoff in present-day simulations because of insufficient precipitation over the
watershed (Fekete et al., 2001; Kelley et al., 2020).

Based on this evaluation of the different ways of translating LPJ PFTs to GISS PFTs, we found
that LtoG_M2 was the scheme that simulates global precipitation and surface temperature most
consistent with observations, and ground albedo that is closest to the standard pre-industrial
boundary conditions dataset used usually used to drive ModelE. Figure 3 shows the difference in
PFT cover fraction using LPJ-LMfire with the LtoG_M2 scheme compared to the standard
ModelE boundary condition land cover data set for the late preindustrial time (PI; 1850 CE).
Compared to the ModelE standard land cover dataset for PI, LPJ-LMfire simulates increased
extent and fraction of most trees (drought broadleaf, evergreen needleleaf, and evergreen
broadleaf). Despite selecting a relatively high threshold for tree height to be classified as shrubs



(up to 11 meters for both arid and cold types) the simulated cover fraction of shrubs is low
compared to the standard PI land cover dataset for ModelE. The coverage of both annual and
perennial $C_3$ grasses is greater in LPJ-LMfire in extratropical and polar regions, similarly, $C_4$
grasses, which are not present in cooler climates, shows greater coverage in LPJ-LMfire in
equatorial regions. LPJ-LMfire simulates some vegetation cover in the Sahara and Arabian
deserts while the standard PI boundary conditions dataset suggests that most of this region is
bare soil.

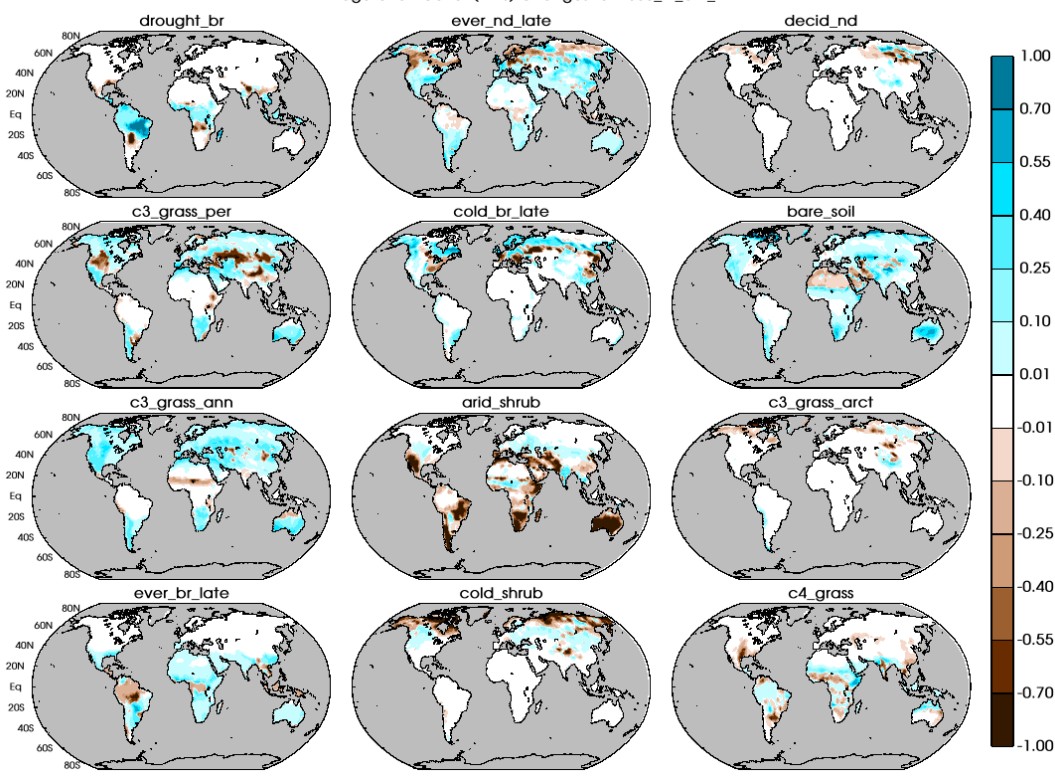


**Figure 3**. Differences between the LPJ-LMfire simulated vegetation distribution (PFTs and land
cover type) and satellite-based land cover boundary conditions used in ModelE for PI control
period under the selected mapping schemes (LtoG_M2).

**3.3 Vegetation Cover Changes under various combinations**





We chose a set of five model configurations (Table 2) to quantify the model bias and interannual
variability in our asynchronous coupling framework for the 2.5ka period. Figures S2.A, S2.B,
S2.C, S2D, and S2.E show the spatial differences between prescribed land cover boundary
conditions maps and land cover interactively simulated by our LPJ-LMfire-ModelE coupled
model, which is henceforth referred to as the "coupled model system". These land cover
difference maps are shown for each of the different model configurations described above,
following the final iteration of the asynchronous coupling when the coupled model system is
assumed to be either equilibrated or the process was truncated due to instability (Table 2).
Figures S2.A, S2.B, and S2.C show the changes in the land cover from the default ModelE land
cover boundary conditions map for PI (Fig S1.A); Figures S2.D and S2.E show the differences
calculated from the modified vegetation following the PMIP4 protocols (Fig S1.B).

Across all configurations, most of the tree PFTs show an increase in cover in the coupled model
system relative to the prescribed land cover maps. However, in simulations where bias correction
to the climate model was not applied, deciduous needleleaf tree cover is reduced in the high
latitudes of the Northern Hemisphere (2.5k_PI_x_x, 2.5k_PI_x_GISS and 2.5k_GS_x_GISS)
and this, in turn, has a substantial impact on regional climate. The coupled model system
simulates increased annual and perennial $C_3$ grass cover across all configurations relative to the
prescribed maps, while the Arctic $C_3$ grass shows a mixed regional response. Increased $C_4$ grass
cover is mostly confined to the equatorial region and Southern Hemisphere; over the Northern
Hemisphere $C_4$ grass cover decreases, irrespective of the inclusion and exclusion of interannual
variability or bias correction. As discussed previously, the extent of arid and cold shrubs is
reduced significantly in the coupled model system relative to the prescribed maps, even when the
threshold height to separate trees shrubs was set at a relatively tall limit of 11 m. A similar
reduction in shrub cover relative to the land cover map used to initialize the simulation
vegetation distributions is also simulated under all configurations.

In Figures 4 and 5 we present heatmap-type diagrams of the mean land cover fraction over
selected regions to demonstrate and understand the pattern of change in vegetation distribution
simulated by the coupled model system. These figures depict changes in land cover under the
different asynchronous coupling experimental configurations used in this study. Vegetation



fraction changes averaged over northern Asia (NAS) (Fig. 4) and eastern Africa (Fig. 5; see Fig.
9 for the region boundaries; NAS: magenta; EAF: blue). Deciduous needleleaf tree cover over
northern Asia (60°N-77°N, 70°E-135°E) is replaced by bare soil in all experimental
configurations where bias correction of the climate model output was not applied. A similar
disappearance of evergreen needleleaf late-successional forests, as well as a quick disappearance
(within the first iteration) of cold shrubs, was also noticed. This suggests that, in the absence of
bias correction the model's drift towards colder conditions strongly influences vegetation growth
in subsequent iterations over higher latitudes, which is inconsistent with the standard land cover
boundary condition dataset used with ModelE (Kelley et al., 2020). On the other hand, when bias
correction is applied along with interannual variability from either model (2.5K_PI_BC_LPJ and
2.5K_GS_BC_GISS), boreal forests are present in the northern Asia region along with cold
shrubs and grasses.

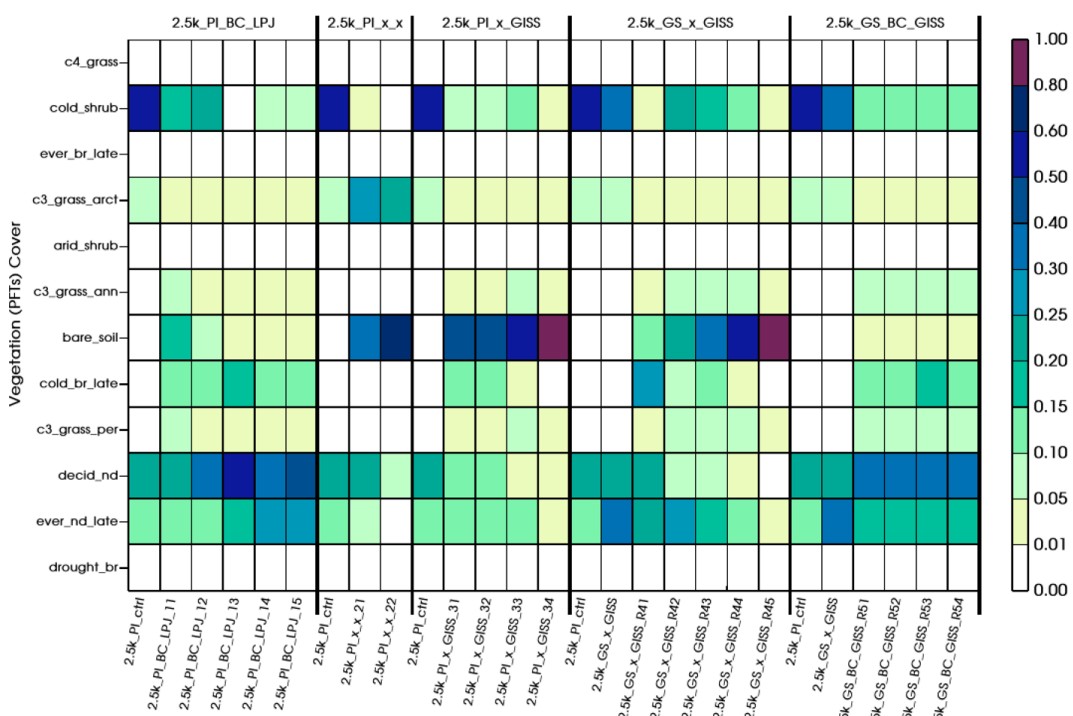






Figure 4. Area average of fractional land cover over Northern Asia (60°N-77°N, 70°E-135°E)
under the range of experimental configurations used in this study.

Over eastern Africa (EAF: 0° N-18° N, 25° E-46° E) the impact of bias correction is less
important than over the high latitudes of the Northern Hemisphere. The presence of broadleaf
tree PFTs (drought broadleaf and evergreen broadleaf) and $C_4$ grasses is consistent across all the
experimental configurations we used. However, the cover fraction arid shrubs decreased
substantially, associated with a slight increase in the bare soil fraction.

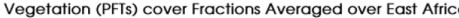

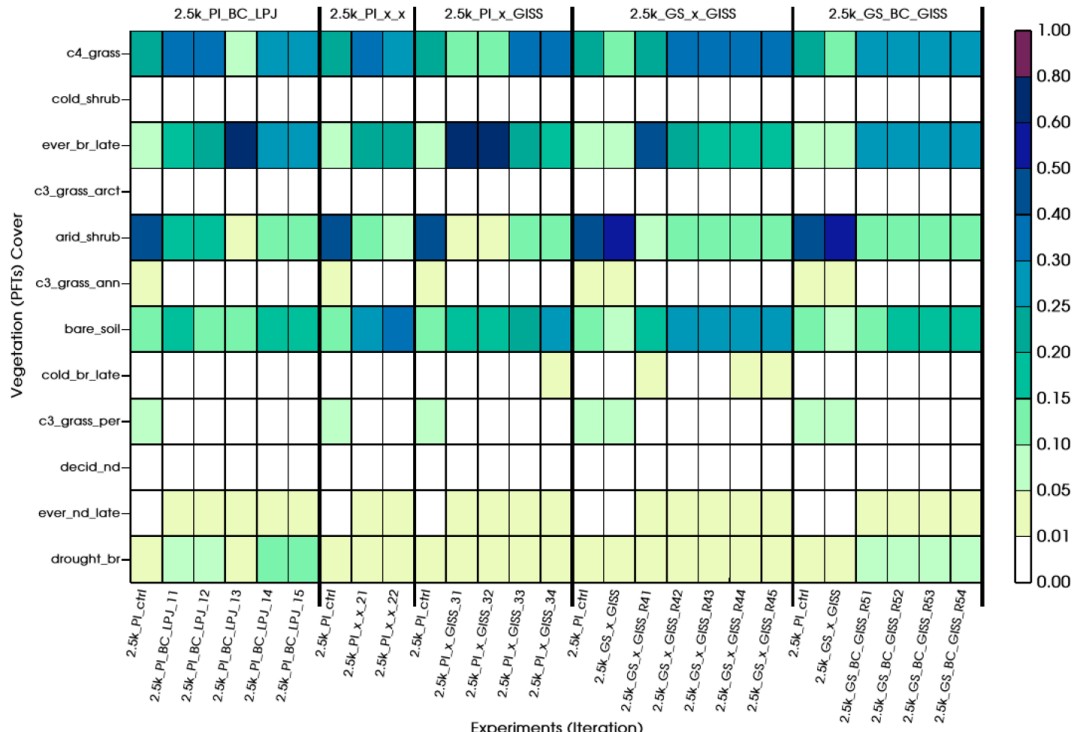


Figure 5. Same as Figure 4A, but for eastern Africa (0°N-18°N, 25°E-46°E).

**4. Global climate response**
To evaluate the spatial features of the equilibrium climate simulated by ModelE, we analyzed the
last 100 years of the final iteration of each coupled model system experimental configuration. We





aimed to understand the biogeophysical feedback due to vegetation cover changes as well as the
role of model configuration on climate. Figure 6 shows surface albedo (%) for ModelE in its initial
PI state, and differences between this initial state and simulated albedo for 2.5ka using the coupled
model system. We used student's t-tests to estimate if the albedo differences were statistically
significant at 95% confidence interval. The coupled model system shows substantial vegetation
cover change over the high latitudes of the Northern Hemisphere. As expected, most of the
significant changes occur over land, while changes in albedo over the oceans are largely
insignificant. The spatial pattern of albedo change differs between simulations where bias
correction was applied (2.5k_PI_BC_LPJ and 2.5k_GS_BC_GISS) and those where it was not
(2.5k_PI_x_x, 2.5k_PI_x_GISS, and 2.5k_GS_x_GISS). Albedo over the high latitudes of the
Northern Hemisphere decreases up to 10% caused by increased tree cover fraction (deciduous
needleleaf and evergreen needleleaf) in the coupled model system relative to standard PI land
cover dataset.

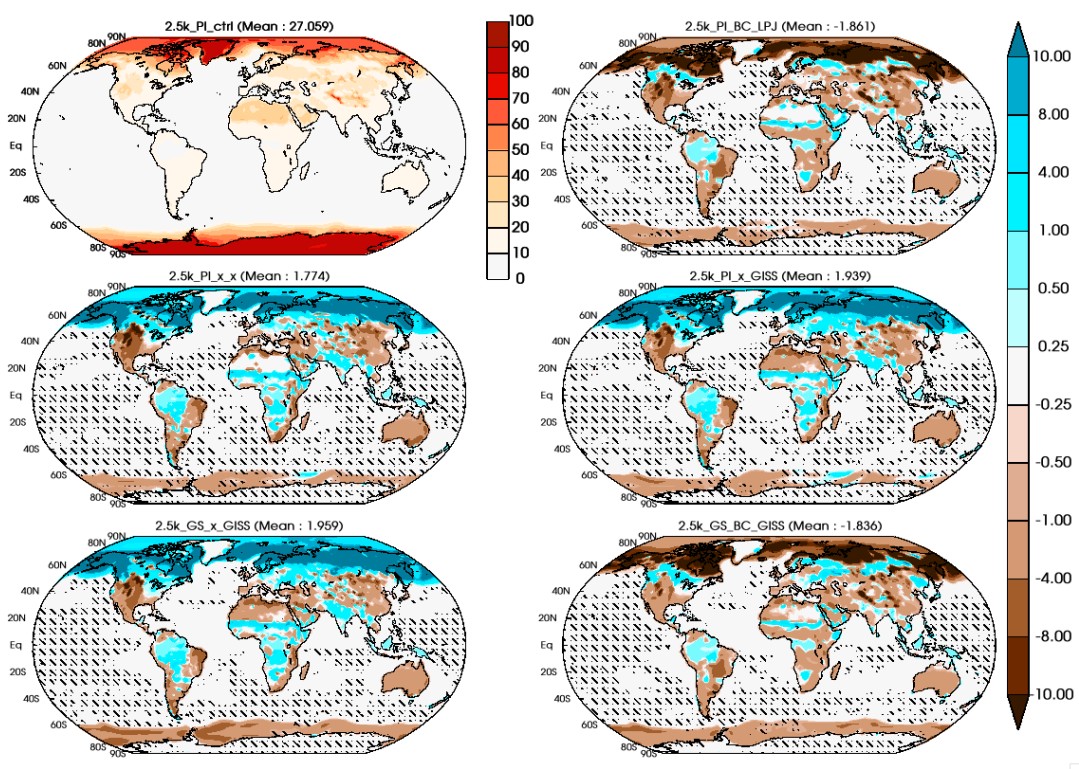






Figure 6. Annual mean (top left; 2.5k_PI_ctrl) and change (all other panels) of surface albedo (%)
for the various configurations listed in Table 2. Stippling indicates the region over which change
is statistically insignificant at a 95% confidence interval (student's t-test).

This increased tree cover fraction subsequently absorbs more incoming solar radiation and raises
surface temperature by 2-4 °C over high latitude regions compared to the control run (Fig. 7 top-
right and bottom-right panels). In experiments where bias correction was not applied
(2.5k_PI_x_x, 2.5k_PI_x_GISS and 2.5k_GS_x_GISS), the relatively cold conditions simulated
by the coupled model system shows an opposite albedo-vegetation response (> 3 °C cooling over
Northern Hemisphere high latitudes). This strong drift towards a colder climate in the absence of
bias correction resulted in the continuous formation of sea ice that ultimately reaches the
(shallow) seabed, effectively creating land ice and eliminating the ocean from the gridcell. In
coupled model system experiments without bias correction, we terminated the iterative processes
when this freezing of the ocean to the seabed occurred, because this condition caused the model
to crash (2.5k_PI_x_x, 2.5k_PI_x_GISS, and 2.5k_GS_x_GISS).

At lower latitudes, albedo tends to show decreases relative to the standard boundary conditions
in all experiments, particularly over the forested areas of the equatorial regions and temperate
latitudes of the Northern Hemisphere. Over the northern Africa and the Indian subcontinent
changes in both albedo and surface temperature are more mixed. Albedo change in central and
northern Africa driven by a reduction in the area occupied by shrubs and an increase in bare soil
fraction. This pattern of increased albedo is more prevalent in simulations that were initialized
with Green Sahara land cover boundary conditions.



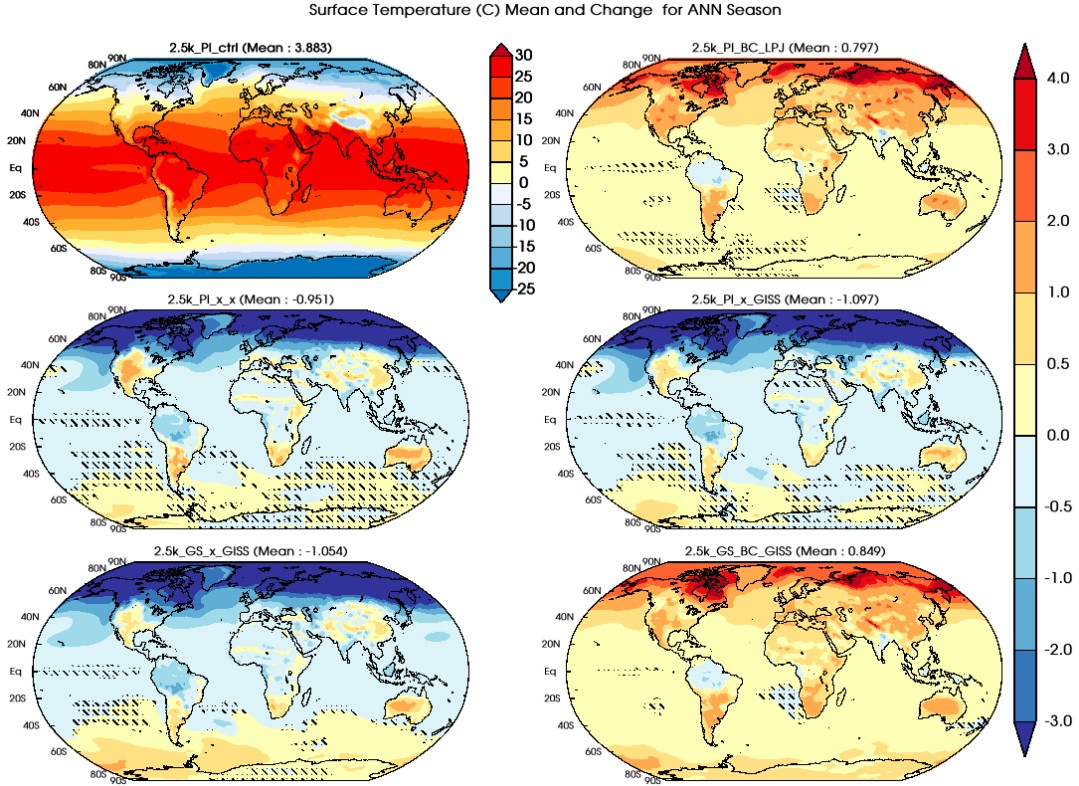


Figure 7. Same as figure 6 for Surface air temperature (°C) mean and change on an annual scale
(ANN season).

In experiments that were initialized with "Green Sahara" land cover boundary conditions where
interannual variability from GISS ModelE is included with and without adopting the bias
correction, comparison of the surface temperature response between simulations with
(2.5k_GS_x_GISS; Figure 7, bottom-left) and without bias correction (2.5k_GS_BC_GISS;
Figure 7, bottom-right) reveal the significance of bias correction for the asynchronous coupling
process. Broadly, we can observe that bias correction induces a warming of 0.7-0.8 °C, and
exclusion leads to a cooling of 0.9-1.1 °C, at the global scale, predominantly over the northern
hemisphere land regions.

Precipitation change across the model configurations is shown for Northern Hemisphere summer
(JJAS) at global scale in Figure 8. The significance of bias correction is noticeable over the high



latitudes of the Northern Hemisphere. Simulations with bias correction (2.5k_PI_BC_LPJ,
2.5k_GS_BC_GISS) lead to an increase in JJAS season precipitation relative to the initial
boundary conditions, while those experiments without bias correction (2.5k_PI_x_x,
2.5k_PI_x_GISS) show reductions in precipitation. Reductions in precipitation relative to initial
conditions are visible in Europe in all configurations and are greater in experiments where bias
correction was not applied. Another common feature among the experiments was the variable
spatial pattern of JJAS precipitation change over tropical regions. All configurations showed
increased precipitation over south and east Asia. Over the Nile headwaters in East Africa
(Melesse et al., 2011) precipitation increased, particularly in those experiments where bias
correction was applied. Interestingly, increased Northern Hemisphere summer monsoon
precipitation season (JJAS) over the Asian continent was simulated across all configurations. In
contrast, only a marginal northward procession of ITCZ over tropical Africa was simulated.

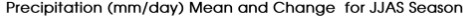

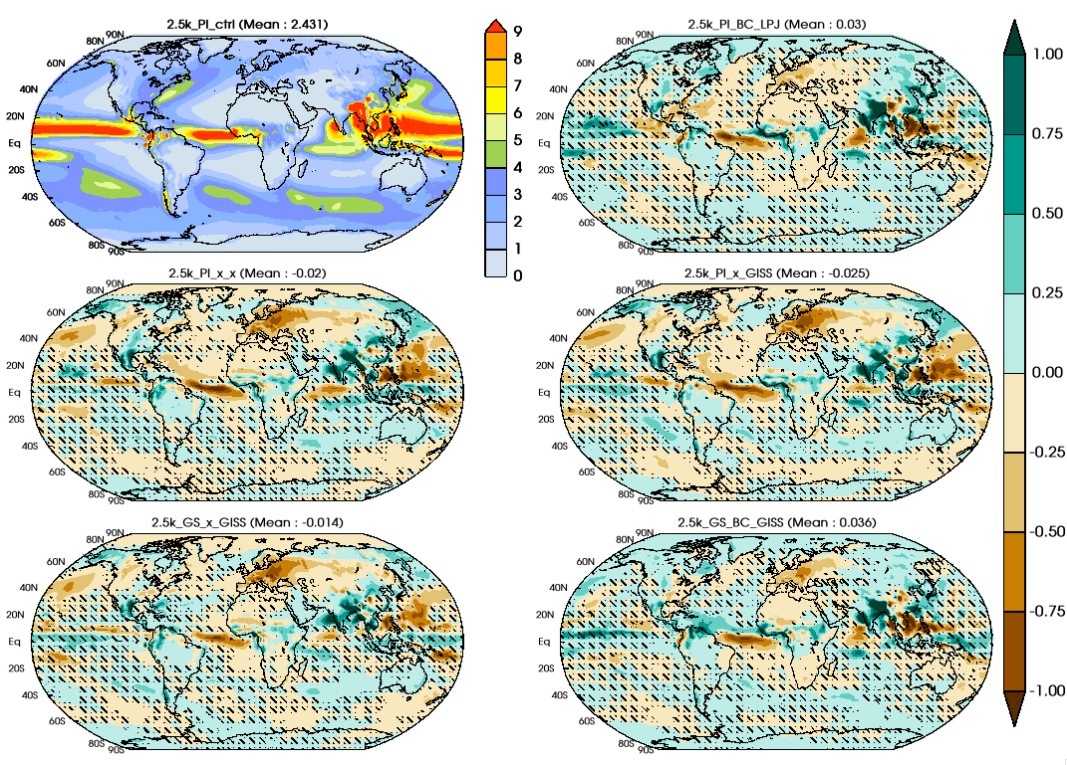





Figure 8. Same as figure 6 for precipitation (mm/day) mean and change on an annual scale (JJAS
season).

## 4.1 Regional climate

The spatial pattern of changes in climatic features for 2.5ka using our coupled model system
shows several prominent and robust regional signatures of climate change. We selected nine
regions over land (Fig. 9; Table 3) to analyze regional temperature and precipitation changes in
our simulations. Area-averaged time-series anomalies with respect to the 2.5ka control run
(2.5k_PI_ctrl) for the various experiments performed are calculated for these different regions.

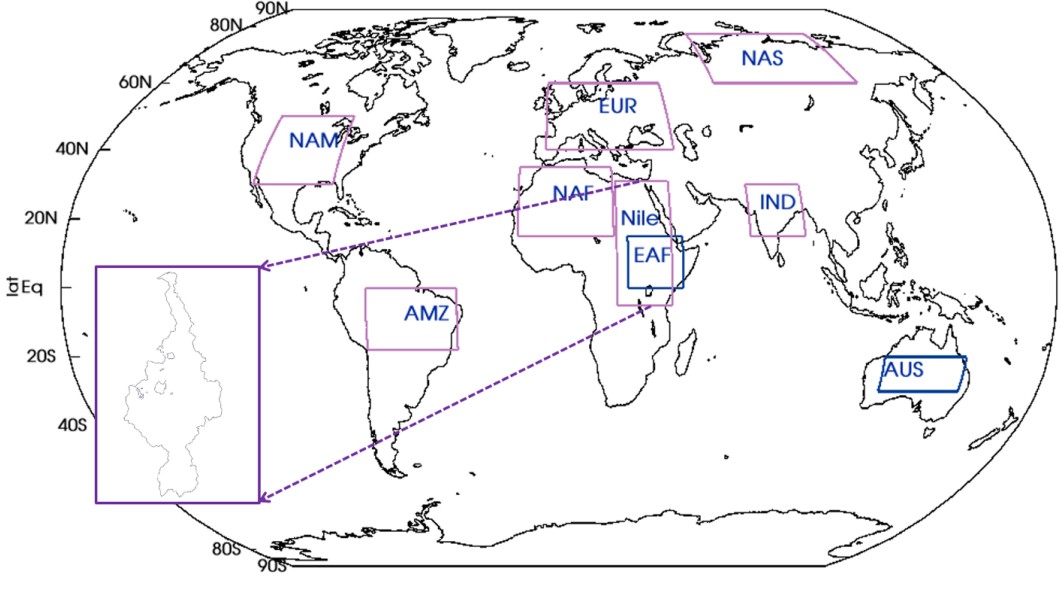



Figure 9. Boundaries for the regions used for regional analysis. The inset map shows the Nile
River basin in high resolution, which is superimposed upon the ModelE resolution to generate
the grid-specific weights for the Nile River basin. The EAF and AUS regions are used in
Figs. 4A and 11.




**Table 3: -** Regions details including the boundary co-ordinates for all the regions.

| Region (long name) | Region (short name) | Region boundary (Latitudes) | Region boundary (Longitudes) |
|---|---|---|---|
| North America | NAM | 30°-50° N | 115°-85° W |
| Amazon Rainforest Region | AMZ | 0°-18° S | 37°-70° W |
| Northern Asia (Siberia) | NAS | 60°-77° N | 70°-135° E |
| North Africa | NAF | 15°-35° N | 15° W-20° E |
| Europe | EUR | 40°-60° N | 5° W-45° E |
| Indian Region | IND | 15°-30° N | 70°-90° E |
| Nile River Basin | Nile | 5° S-31° N | 21°-41° E |
| East Africa | EAF | 5°-15° N | 25°-45° E |
| Australia | AUS | 20°-30° S | 120°-150° E |


Figure 10 shows box-and-whisker plots of mean and median annual surface temperature (top)
and JJAS seasonal precipitation (bottom) change, as well as the 5-95 percentile range along with
the upper and lower quartiles (25th and 75th percentiles) of the anomaly time series for each
region. As suggested from the global analyses of spatial patterns, the shift towards relatively
warmer or colder climate as a result of applying bias correction is evident. Bias correction leads
to strong warming over northern Asia (NAS region) of 3-4 °C, while without bias correction this
region cools by 5-6 °C. The partition between experiments with and without bias correction is
also apparent over selected regions of the mid-latitudes between 35°-60° N (NAS and EUP).

Except for northern Asia (NAS), all regions show approximately similar interannual variability
in mean annual surface temperature. In northern Asia interannual variability is greater, especially
in simulations where bias correction was not applied. Our results show that interannual
variability in summer temperature in northern Asia is sensitive to changes in land cover, with
greater variability in simulations where bias correction was not applied.



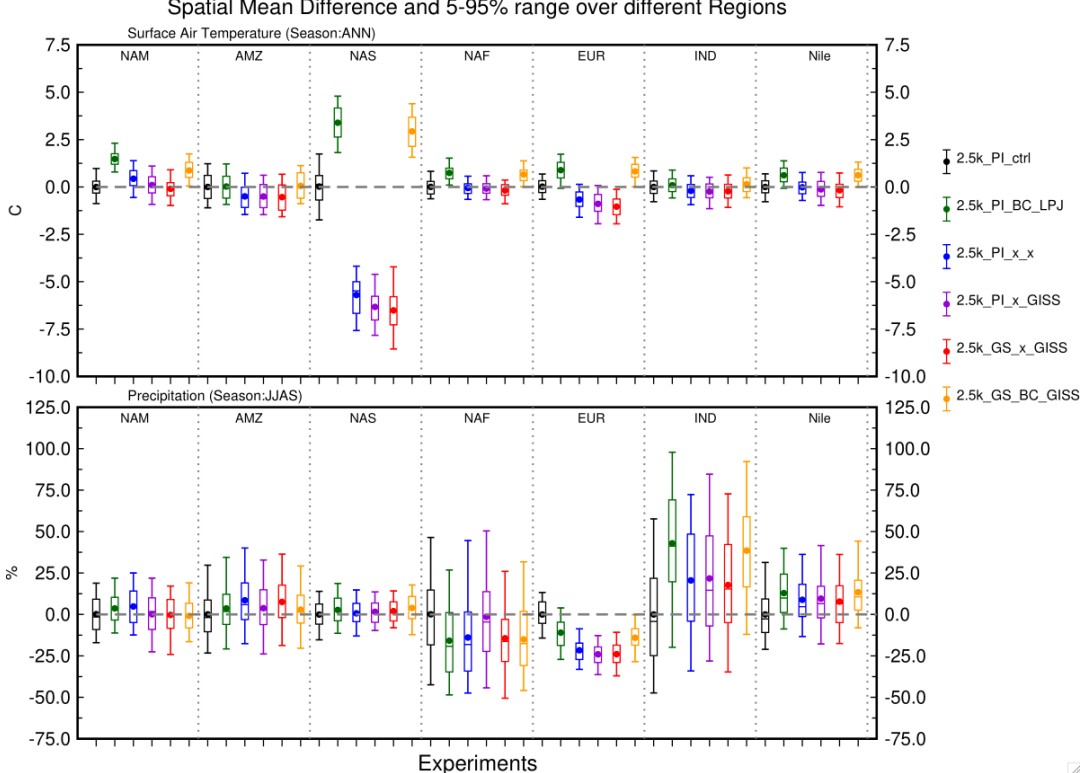


Figure 10. Regional change in surface air temperature (top panel, °C, annual mean) and
precipitation (bottom panel, %, JJAS) for the various simulations with respect to the 2.5ka control
run (2.5k_PI_ctrl). Regions name as listed in table 3.

Simulated 2.5ka precipitation for the Northern Hemisphere summer (JJAS) shows substantial
changes in mean state relative to the 2.5ka control with PI vegetations, particularly for the
tropical regions of northern Africa, India, and the Nile basin (Fig. 10, bottom panel). Interannual
variability in precipitation is comparable to the initial control run (black line). However, the
magnitude of variability differs across the regions; it is more prominent in tropical regions than
in the extratropics. An increase in mean precipitation of order of 20-30% without bias correction
and up to 40% with bias correction is simulated in JJAS season precipitation for the Indian
summer monsoon region (IND and it is in a range of 10-25% increase over the Nile basin region.
A drying pattern over Europe (EUR) ranges from 10-25% and is consistent for all the
simulations; a greater decrease in European precipitation was simulated when bias correction is





not adopted. A similar drying pattern was also simulated over the North America (NAM) and
northern Africa (NAF) regions. The relatively small magnitude of interannual variability in
precipitation over Europe and North America suggests that model does not produce high
variability across these regions and that it is not sensitive to the different experimental
configurations. Despite the large changes in both mean state and variability in temperature,
precipitation over northern Asia (NAS) changes little from the control state and across
simulations. In the Amazon region (AMZ), precipitation changes were small and not
significantly different between simulations. Without bias correction, the coupled model system
suggests a modest increase in mean seasonal precipitation up to 10%. We also noticed a similar
response of slightly increased precipitation in Southern Hemisphere summer (DJF) over
Australia (not shown here).

We further investigated the way our experiments influenced the seasonal cycle of temperature
and precipitation over the regions discussed above. Our results show that the seasonal cycle of
surface temperature is broadly similar across experiments for all the equatorial regions except the
Amazon (AMZ) region, where surface temperature is reduced by 0.5 °C in experiments where
bias correction was not applied (Fig. S3). Over the northern Asia (NAS) region, we see a
considerable difference in the seasonal cycle of temperature of 5-15 °C between runs with and
without bias correction. The seasonal cycle of temperature in the 2.5ka control (2.5k_PI_ctrl)
simulation over NAS is intermediate to the experiments but tracks closer to the simulations
where bias correction was applied, particularly in Northern Hemisphere winter, where, as noted
above, simulations without bias correction result in very cold conditions in this region.



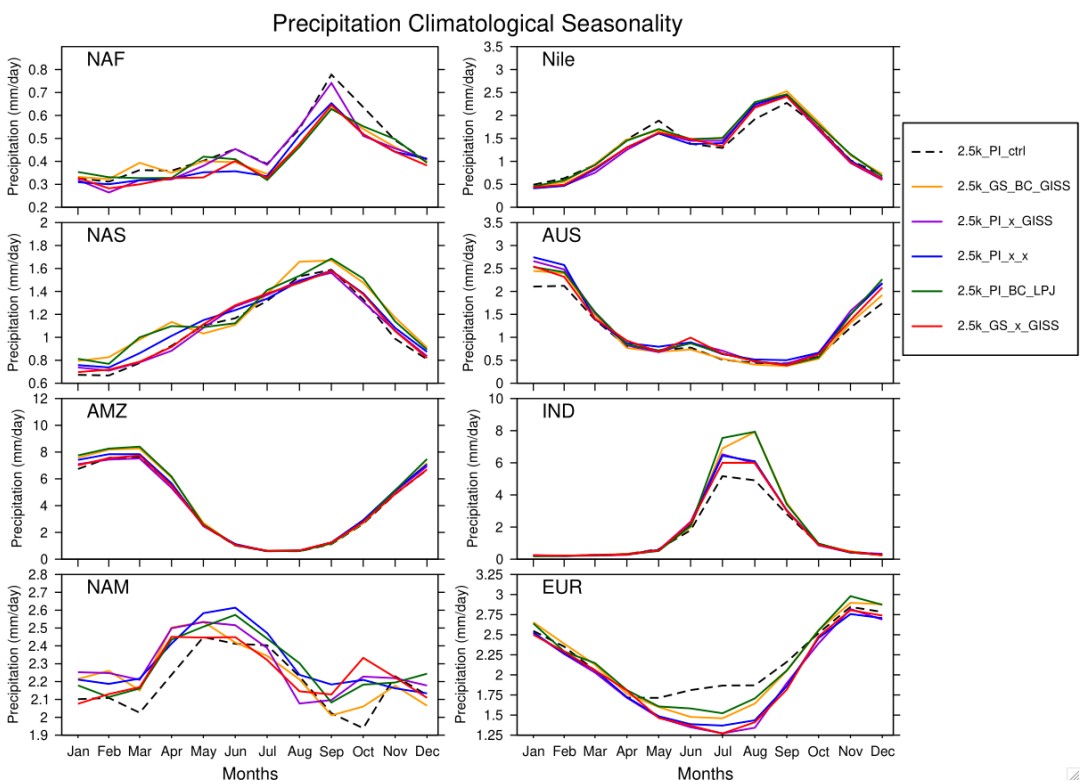


Figure 11. Seasonality of precipitation averaged over the selected regions.

Compared to temperature, the seasonal cycle of precipitation shows greater differences among
simulations over several of the regions (Fig. 11). An increase of 2-3 mm/day over the Indian
region (IND) is simulated during the Indian Summer Monsoon months (JJAS) when using LPJ-
LMfire-generated land cover for both types of experiments (with and without bias correction),
with the bias-corrected simulations showing a larger increase in precipitation than the non-bias-
corrected ones. When bias correction is applied, the seasonal peak of precipitation shifts from
July to August. Over Europe, we observe a decrease of up to 0.5 mm/day in summer
precipitation relative to the control simulation in all simulations that use the LPJ-LMfire PFTs.
Precipitation decreases even more when the bias correction was not applied. The North Africa
region (NAF) also shows a slight decrease in precipitation relative to the control over most of the
seasonal cycle, while in North America (NAM) we see an increase in precipitation outside of the
JJAS summer months. The Amazon rainforest region (AMZ) shows no change in the seasonal
cycle of precipitation in all experiments. The Nile River basin (Nile) and Australian (AUS)





regions also show small increases in precipitation relative to the control in their respective
monsoon seasons (JJAS and DJF).

## 5.0 Comparison with paleoclimate-proxy records for 2.5ka

To evaluate the coupled model system's skill in representing past climate, we compared our
simulations for 2.5ka with multiproxy temperature reconstructions and speleothem-based oxygen
isotope records.

### 5.1 Comparisons with reconstructed temperature

Kaufman et al. (2020) used five different statistical methods to reconstruct temperature at 1319
globally distributed sites covering part or all or Holocene from a range of proxy types. For each
method, a 500-member ensemble of plausible reconstructions was presented. For comparison
with our model output, we extracted temperature anomalies for 2.5ka (relative to the value
reconstructed for the late preindustrial Holocene) from the ensemble reconstructions which we
binned into six latitude bands between the North and South Poles (each 30 degrees wide). We
computed the mean and median zonal anomaly using all 500 estimates of mean surface
temperature (MST) over each band for each of the five methodologies (total 2500), along with
the 5-95 percentile interval to represent uncertainty/variability among the sites in the zone and
across reconstruction methods (black bar in Figure 12) as suggested (Kaufman et al. 2020).



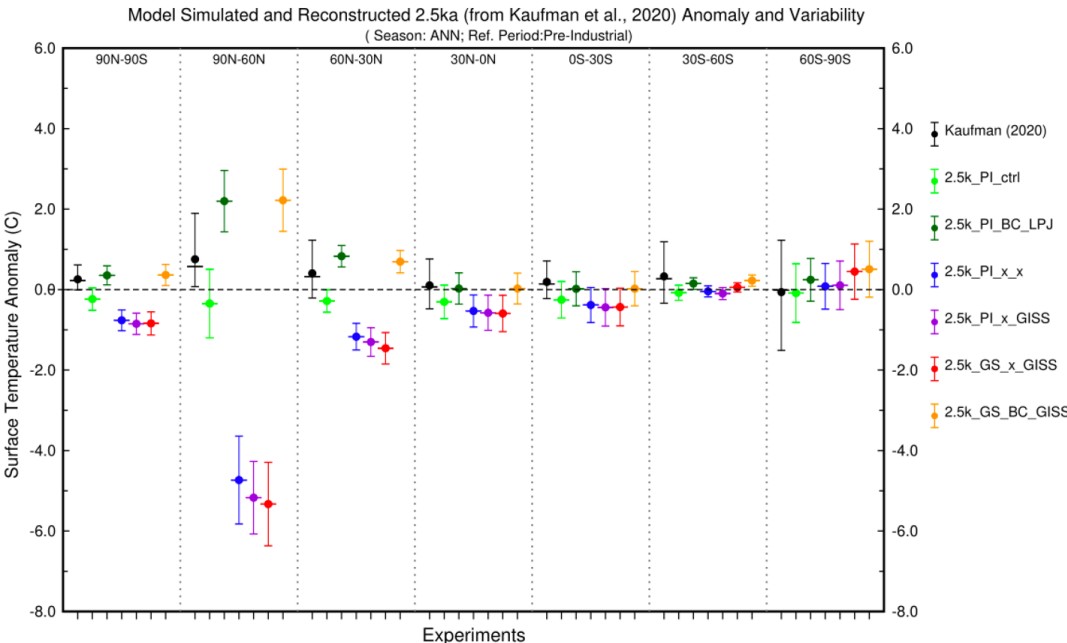


Figure 12: Comparison of model simulated annual surface temperature anomalies and interannual
variability for 2.5ka (with LPJ-LMfire vegetation) against the independent proxy-based
temperature reconstructions (black, Kaufman et al., 2020). Mean (circle), median (line) along with
5-95 percentile range as variability bars (whiskers) and different colors represent our different
experiments.

On global mean and in all latitude bands except the most southern one, proxy reconstructed
surface temperature is slightly warmer at 2.5ka relative to the late preindustrial. Model
simulations where bias correction was not applied show colder conditions than the
reconstructions globally and in the Northern Hemisphere. These differences between model and
proxy are very large in the high latitudes of the Northern Hemisphere and statistically significant
throughout the extra-tropics. In the Southern Hemisphere, the differences between model and
proxy reconstructions are smaller and insignificant, and there is less difference between
simulations with and without bias correction. It should be noted that the larger uncertainty in
reconstructed temperature over the southern polar band is due to a noticeably lower number of
available proxy records (157 records; Kaufman et al., 2020).



**5.2 Comparisons with speleothem oxygen isotope ratios**
ModelE2.1 includes a representation of the stable water isotopologues as passive tracers and the
isotopic composition of precipitation can be diagnosed from the model output. We compared the
simulated mean annual isotopic composition of precipitation ($\delta^{18}O_p$) with oxygen isotope records
from the Speleothem Isotope Synthesis and Analysis (SISAL) version 2 database (Comas-Bru et
al., 2020). Using the published chronologies for each speleothem record we extracted all samples
dated between 3-2 ka, which resulted in 163 measurements from 111 sites. Depending on their
mineralogy (i.e., calcite or aragonite), the mean $\delta^{18}O$ values (VPDB) were converted to their drip
water equivalents that could be compared to simulated $\delta^{18}O_p$ (VSMOW) (Comas-Bru et al.,
2020). We used simulated mean surface air temperature obtained from the grid points nearest
each cave sites to estimate the cave temperature required to convert mineral $\delta^{18}O$ to an
equivalent the drip water value. For each of our model experiments, we extracted simulated
$\delta^{18}O_p$ nearest to each cave site and compared it with the estimated drip-water $\delta^{18}O$.
Overall, the mean $\delta^{18}O_p$ spatial distribution in all 2.5ka simulations is in excellent agreement
with the proxies, showing better pattern correlations ($r_{pat}$) than 0.83 (Figure 13), with
the 2.5k_PI_x_x iteration marginally showing the highest skill (i.e., $r_{pat}$ = 0.85 and RMSE =
1.90; shown in supplementary Fig S4). For comparison, the worst simulation using this metric,
2.5k_GS_BC_GISS, is almost as equally skillful ($r_{pat}$ = 0.84 and RMSE = 1.92; Fig. S4),
demonstrating that none of the different configurations we presented here were significantly
different.

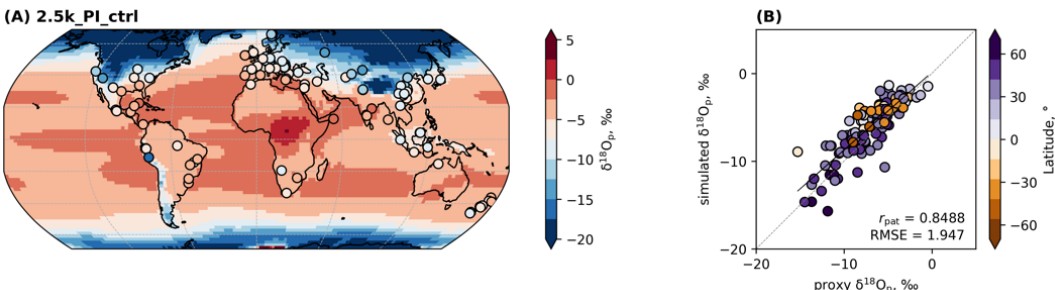


Figure 13. Comparison of simulated $\delta^{18}O_p$ with speleothem $\delta^{18}O$. Left: global distribution (70° S-
70° N) of simulated $\delta^{18}O_p$ (background) and speleothem $\delta^{18}O$ (circles), converted to their drip
water equivalents (see text) for the 2.5k_PI_ctrl simulation. Right: scatterplots between simulated
and proxy $\delta^{18}O_p$. Black line represents the least squares regression fits to data points while the gray





dashed line represents the 1:1 line. $r_{pat}$ and RMSE are reported in the lower right corner of the
scatterplot. For comparison against each model experiment, see Fig. S4

Regionally, we similarly found that most simulations show no significant deviation with each
other (Figure 14, Figure 15). We note, however, that over Europe (Figure 15E), variability may
be explained by the observed change in magnitude on both SAT and summer precipitation
among simulations (Figure 7, 8, 10). Over India and Central Asia (Figure 15F), simulations with
bias correction show lower correlation and higher RMSE values compared to other models
against proxy $\delta^{18}O_p$. This is likely related to the observed increase in mean summer precipitation
over this region (Figure 10) that were not reflected in the proxy sites.

Compared to proxy $\delta^{18}O_p$, simulations over certain regions show better agreement. Europe,
which is the most densely sampled region, show the best agreement with the proxies (i.e., high
correlation, closest to the reference point, Figure 15E) with the 2.5k_PI_x_GISS iteration best
capturing the spatial $\delta^{18}O_p$ pattern (i.e., $r_{pat} = 0.94$ and RMSE = 1.26). In contrast, simulations
over Central America, South America and Africa show the least skill where the magnitude of
$\delta^{18}O_p$ change are consistently underestimated (i.e., moderate to high correlation but farthest away
from the reference point).  This may largely be due to inadequate sampling in these regions,
especially for Africa, and/or both precipitation and SAT influencing $\delta^{18}O$ may be underestimated
at these proxy locations, resulting in a generally muted $\delta^{18}O$ response across simulations. Cave-
specific factors that alter speleothem $\delta^{18}O$ (e.g., groundwater mixing, fractionation, (Baker et al.,
2019; Hartmann and Baker, 2017; Lachniet, 2009) are also not effectively reproduced in the
models, contributing to the proxy-model mismatch. Regions where the largest simulated SAT,
precipitation, and $\delta^{18}O_p$ change relative to the 2.5k_PI_ctrl are observed, such as northern Africa,
the Amazon basin and Siberia, are not adequately represented by reconstructions, highlighting
the need to expand the proxy network to marine-based records and polar regions over the period
of interest to capture the full range of isotopic variation.



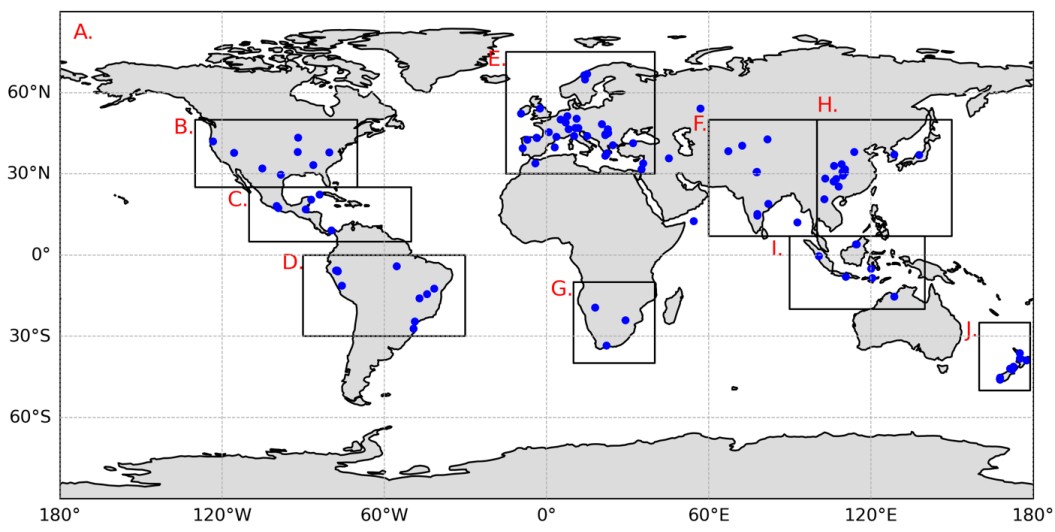

Figure 14. Demarcation of each geographical region. Labels A to J correspond to the respective Taylor diagram plots in Figure 15.










Figure 15. Taylor diagrams showing the r, SD and RMSE values between the proxy-derived and
simulated $\delta^{18}O_p$ for each 2.5k iteration globally (A) and at each subregion (B to J). Subregions are
demarcated in supporting figure 14.
**6.0 Discussion and Conclusions**
Here we presented a generalized technical framework for asynchronously coupling a climate
model (NASA GISS ModelE2.1) with a dynamic vegetation model (LPJ-LMfire) i.e., the "coupled
model system", and demonstrate its skill in reconstructing climate in the late preindustrial
Holocene and for 2.5ka. We examined the role of bias and interannual variability corrections in
this process, and showed how they influence simulated land cover and climate. We demonstrated
the importance of considering such metrics in such a framework in our experimental design and
global and regional scale analyses. We performed a detailed evaluation and comparison of the
climate simulated by the coupled model system with reconstructions of air temperature (Kaufman
et al., 2020) and the isotopic composition of precipitation ($\delta^{18}O_p$) based on speleothems (Comas-
Bru et al., 2020). Similarly to previous studies that used asynchronous coupling to simulate
regional and global paleoclimate ( Kjellstrom et al., 2008; Texier et al., 1997; Noblet et al., 1997;
Velasquez et al., 2021; Claussen, 2009; Strandberg et al., 2011, 2014), we assessed the influence
of the biogeophysical feedback between land and atmosphere.
Our results demonstrate the strong influence of including bias correction when passing simulated
climate to the land surface model. To correct biases inherent in the climate model, in selected
experiments we passed climate anomalies relative to a control simulation to the land model that
were added to a standard baseline climatology based on contemporary observations. In simulations
without this bias correction, raw simulated climate was passed directly from ModelE to LPJ-
LMfire. Where bias correction was applied ModelE drifts towards warmer climate; simulations
without bias correction drift towards colder climate. This effect was especially apparent in the high
latitudes of the Northern Hemisphere, particularly over Asia. With bias correction, high latitude
vegetation is dominated by tree plant functional types, while without it, cold shrubs and arctic
grasses are the predominant form of land cover. These results are characteristic of the well-known
vegetation-albedo feedback that is important at high latitudes (Charney et al., 1977; Charney,
1975; Doughty et al., 2012, 2018; Pang et al., 2022; Stocker et al., 2013; Swann et al., 2010; Zeng
et al., 2021).

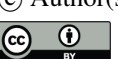



The effects of bias correction on precipitation were less apparent and confined to regional scale.
We simulated a greater Indian summer monsoon season (JJAS) precipitation with bias correction
(>1 mm/day), and a nominal increase of ~0.5 mm/day across east China, Africa, and the North
American monsoon region. In other regions, the patterns of precipitation change were similar
across all experiments except for Europe where drier conditions are simulated in summer (up to –
1 mm/day) in simulations where bias correction was not applied.

The high latitudes of the Northern Hemisphere were also the region with the largest disagreement
between model and independent, multi-proxy temperature reconstructions. These comparisons
also highlighted the important role of bias correction; experiments with correction were much more
similar to reconstructions than those without. Simulations of the isotopic composition of
precipitation ($\delta^{18}O_P$) shows an excellent agreement with speleothem records with a pattern
correlation greater than 0.8. However, the difference in the magnitude of model simulated $\delta^{18}O_P$
from proxies over various regions indicates an underestimation of relationship between surface
temperature and $\delta^{18}O_P$ variability (Henderson et al., 2006; Kurita et al., 2004). A global evaluation
of model skill is hindered by the difference in the number of independent paleoclimate
reconstructions available for different regions, particularly in north Asia where we see the greatest
sensitivity of the coupled model system to the experimental setup. When examining modeled and
reconstructed $\delta^{18}O_P$, in Europe, which is the region with the greatest number of records, we see a
stronger pattern correlation with lower RMS values as compared to other regions.

In this study, we confirmed the importance of the land surface for simulating paleoclimate, even
for the late Holocene where land surface conditions were not as different from present as they were
during, e.g., the last glacial cycle or even mid-Holocene. We demonstrated that asynchronous
coupling can be a computationally inexpensive way of capturing land-atmosphere feedbacks and
improving the fidelity of the simulated climate. We noted that correcting bias present in the climate
model is essential for simulating climate that is consistent with independent reconstructions,
particularly for the high latitudes of the Northern Hemisphere. Future work with the coupled model
system will include quantification of the influence of major volcanic eruptions for regional and
global paleoclimate (Singh et al., 2024, in preparation) and the influence of past climate on the
dynamics of complex civilizations in prehistory.





**Code/Data availability**

Details to support the results in the manuscript is available as supplementary information is provided with the manuscript. GISS Model code snapshots are available at https://simplex.giss.nasa.gov/snapshots/ (National Aeronautics and Space Administration, 2024), LPJ-LMFire (https://zenodo.org/records/5831747), and important codes, calculated diagnostics as well as other relevant details are available at zenodo repository (https://doi.org/10.5281/zenodo.13626434) (Singh et al., 2024). However, raw model outputs data and codes are available on request from author due to large data volume.

**Acknowledgements**

RS, KT, ANL and FL acknowledge support by the National Science Foundation under Grant No. ICER-1824770. ANL acknowledges institutional support from NASA GISS. Resources supporting this work were provided by the NASA High-End Computing (HEC) Program through the NASA Center for Climate Simulation (NCCS) at Goddard Space Flight Center and from the Department of Earth Sciences at The University of Hong Kong. The authors thank for their input through multiple discussions the project members and collaborators of the ICER-1824770 project, 'Volcanism, Hydrology and Social Conflict: Lessons from Hellenistic and Roman-Era Egypt and Mesopotamia'. RS and FL acknowledge additional support from European Research Council grant agreement no. 951649 (4-OCEANS project).

**Author's contributions**

RS, KT and ANL identified the study period in consultation with the other authors and RS, AK, KT, ANL and JOK designed the asynchronous coupling framework. RS and AK implemented it and performed the simulations using NASA GISS ModelE and LPJ-LMfire models. IA and RR provided the essential technical support while implementing the framework. RS and RDR created the figures in close collaboration with KT, ANL. RS wrote the first draft of the manuscript and RDR, KT, ANL, and JOK led the writing of subsequent drafts. All authors contributed to the interpretation of results and the drafting of the text.

**Competing interests**

The authors declare no competing interests.



**Short Summary**


This study presents and demonstrates an experimental framework for asynchronous land-
atmosphere coupling using the NASA GISS ModelE and LPJ-LMfire models for the 2.5ka period.
This framework addresses the limitation of NASA ModelE, which does not have a fully dynamic
vegetation model component. It also shows the role of model performance metrics, such as model
bias and variability, and the simulated climate is evaluated against the multi-proxy paleoclimate
reconstructions for the 2.5ka climate.

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
