# Peer review of "Modelling framework for asynchronous land-atmosphere coupling using"

_Geoscientific Model Development, 2024_

## Author Comment (AC1)

We would like to thank both reviewers for their comments that helped improve our manuscript. Please see below replies to all comments. The reviewers' (RC1) comments are in black, our replies in blue, and any updated or new text from the revised manuscript in "*quoted blue italics*". All line numbers mentioned below are from the submitted manuscript.

**Reviewer 1**

Singh et al. "Modelling framework for asynchronous land-atmosphere coupling using NASA GISS ModelE and LPJ-LMfire: Design, Application and Evaluation for the 2.5ka period"

GENERAL

This study simulates the climate of the 2.5 ka period using the NASA GISS ModelE, which is asynchronously coupled to the dynamic vegetation model LPJ-LMfire. The authors conducted several sensitivity experiments to assess the impacts of initial land cover and bias correction on paleoclimate and ancient vegetation. Additionally, they used multiple proxy datasets to validate the simulation results. While this study represents a considerable effort in developing a modeling framework and conducting numerical simulations, the robustness of the main results is questionable, and the study's novelty remains unclear.

First, the study lacks sufficient novelty. The primary focus is on asynchronous land-atmosphere coupling between a climate model and a vegetation model. However, it is unclear why this coupling approach warrants extensive application, as it does not appear to enhance the simulated paleoclimate. As shown in Fig. 12, temperature anomalies exhibited the lowest biases in the 2.5k_PI_ctrl simulation, which did not include asynchronous land-atmosphere coupling. This finding suggests that vegetation coupling has a minimal impact on improving model performance, thereby weakening the study's novelty.

The novelty of this modeling framework is that this methodology represents a pragmatic approach for simulating the past climate conditions using the climate models without having the dynamic vegetation components, which is different from what we and other such models are currently following under PMIP4. In this study, we applied this methodology for the 2.5ka period, which is not among the standard periods selected for paleoclimate simulations under PMIP4, but crucial for the emergence of several complex societies and their climate interactions. Testing this for the 2.5ka period explicitly demonstrated a very broad and generic applicability of this framework, which can be used either by us for another period, or other modelers, since we describe in detail the steps to be taken and the metrics to be used to objectively quantify when equilibrium has been achieved. Along with highlighting the novelty at several places, a sentence is added into the abstract:

"*This study also presents a framework for incorporating biogeophysical responses into climate models without dynamic vegetation, for simulating past climates, in line with the recommendations of the Paleoclimate Modelling Intercomparison Project (PMIP).*"

The explanation for figure 12 was incomplete, and that might have created confusion, we apologise for that. We modified the figure and the discussion around it, so that the reader will not miss the main points behind it. Please see the detailed changes we made below, where we replied to the specific question about the figure 12 (figure 6 in revised, as per reviewer2's suggestion to move this section earlier). The key point that the reviewer missed because of our presentation of that figure is that the runs with the updated vegetation are actually better than the standard preindustrial one (lowest biases) when compared against reconstructions.

Second, the key sources of discrepancies remain unexplored. While the authors compared results from multiple experiments, they primarily noted that bias correction led to a warmer and more stabilized climate, without further investigating the underlying causes of climate variations across different simulations. In particular, the influence of initial vegetation cover, climate variability, and climate-vegetation interactions on the simulated paleoclimate should be explicitly quantified and explained.

We thank the reviewer for highlighting the need for greater clarity in presenting the results. Accordingly, we have revised the paragraph (L626–L635) by adding the following sentence at its beginning.

*"It shows that the 2.5 ka control simulation with present-day vegetation is comparable to pre-industrial conditions, exhibiting a slightly cooler climate. In contrast, proxy-based surface temperature reconstructions (Kaufman et al., 2020) indicate slightly warmer conditions at global mean as well as across most latitude bands, except the far south (60S-90S). Applying bias correction allows the model to reproduce the same anomaly sign as the reconstruction, with minimal global (90N-90S) mean bias relative to the proxy data (2.5k_PI_BC_LPJ and 2.5k_GS_BC_GISS). Although the magnitude of warming remains higher at the northern hemisphere high latitudes, this framework demonstrates improved capability for incorporating biogeophysical effects of past vegetation by adopting bias correction."*

Furthermore, the δ18O validations revealed similar biases across different experimental configurations (Fig. 15), raising the question: what is the added value of employing different model configurations?

Figure 15 (figure 9 after revisions) illustrates that, despite the high sensitivity of the modeling framework to various factors, The model successfully captures key regional climate features across different areas. This highlights the need for additional paleoclimate proxy reconstructions to enable a more robust evaluation of the model's skill in simulating past regional climates. (See the conclusion section, lines 731–743.)

SPECIFIC

Line 40: "The coupled model system is sensitive to the representation of shrubs." What is the underlying reason for this particular sensitivity? Is this a characteristic of this specific model, or is it a general feature of all coupled models?

This is one of key sensitivity features noticed from the NASA-GISS ModelE-LPJ-LMfire coupling framework and reported as an abstract point of this study. The underlying reason is that the alteration of arid shrubs (over Northern African Region) and cold shrubs (Northern hemisphere high latitudes) with bare soil and trees, respectively, changed the surface albedo, and, consequently, the vegetation-albedo feedback substantially modifies the regional climate  (Charney et al., 1977; Charney, 1975; Doughty et al., 2012; Stocker et al., 2013).

We found this being a characteristic of the particular coupling between GISS ModelE and LPJ-LMfire, and it is a feature of ModelE. We cannot generalize this conclusion to other models.

We modified the sentence as given below:

*"The NASA GISS ModelE found to be particularly sensitive to the representation of shrubs, implying that this land cover type requires particular attention as a potentially important driver of climate in regions where shrubs are abundant."*

Lines 176-177: How well does the LPJ model perform in simulating present-day vegetation? It would be more appropriate to validate the LPJ model using observed meteorological data and vegetation parameters rather than relying solely on PI simulations.

These lines in the text do not refer to an LPJ simulation but rather the boundary conditions used to force the initial (0th order; 2.5ka) simulation of ModelE. Nevertheless, LPJ has been shown to simulate realistic global distributions or present-day biogeography (Sitch et al., 2003) and wildfire (Pfeiffer et al., 2013; Thonicke et al., 2010). In the description of LPJ-LMfire (section 2.1.2) we added a few words to emphasize that the model has been evaluated in the context of contemporary observations.

*"LPJ-LMfire has been successfully validated for simulating present-day biogeography and fire regime characteristics, and its outputs have been compared against contemporary observations (Sitch et al., 2003; Thonicke et al., 2010; Pfeiffer et al., 2013)."*

Line 207: What observational datasets could be used to validate the derived 'wet days? Additionally, models tend to overestimate the frequency of small rainfall events, which suggests that the nonlinear relationship derived from observations may not be directly applicable to model outputs.

We thank the reviewer for this insightful comment. It is precisely because climate models tend to overestimate the frequency of small rainfall events (positive drizzle bias) that instead of using daily (or 6-hourly) precipitation from the ModelE output to provide the wet days variable required by LPJ-LMfire, we use a dataset based on daily observations of precipitation (CRU TS). Since this observational dataset is not available for periods in the past, we use the local regression model as an empirical, observation-based way of estimating wet days under paleoclimate conditions. This simple relationship does not capture non-linearities, but in the absence of past precipitation data we can't know the extent of such non-linearities, let alone include them in our calculations.

Line 220: "Adding interannual variability"—why is this an important consideration? How do simulations differ with and without interannual variability?

We added the following in the text in section 2.3.3:

*"LPJ requires climate input data with interannual variability because fires and other disturbance events occur only in years with anomalous climate, for example hot or dry years Sitch et al. (2003). Driving the model with a climatological mean climate will result in disturbance frequencies that are lower than the expected mean, that in some regions would lead to an overabundance of tree cover when we would expect herbaceous vegetation."*

Also, since the ModelE simulations presented here include a dynamic ocean, and interannual variability such as the El Niño–Southern Oscillation has a significant impact on global temperature and precipitation, excluding this variability would bias and distort our results.

Line 235: What is the connection between plant functional types (PFTs) and fire frequency? Please clarify the underlying mechanism.

High fire return interval will favor early successional plant functional types because the elapsed time since the last disturbance would not have been long enough for the establishment and/or dominance of late successional plant functional types. By definition, late-successional plant functional types require long periods of low disturbance to be present in ecosystems.

We have added a sentence to section 2.3.3 to clarify this point.

*"High fire frequency favors early-successional PFTs because the time between disturbances is shorter than that required for establishment. By definition, late-successional PFTs require extended periods of low disturbance to persist within the ecosystem"*

Line 243: "A predefined threshold"—Is this threshold applied uniformly at the global scale, or do different grid cells use varied thresholds? How was this threshold determined?

Yes, this classification is applied uniformly over the global scale. We modified this sentence: "A *globally-uniform* predefined threshold".

Tree growth is one of fundamental and widely adopted criteria for vegetation models to distinguish different PFT types (forest trees, shrubs, herbs) (DeFries et al., 1995; Kim et al., 2015)(DeFries et al., 1995; Kim et al., 2015). Thus, we designed and evaluated a set of thresholds here as detailed in table S1, which is already cited in the manuscript.

Line 272: "climate vegetation models"—this phrase is missing an "and" (should be "climate and vegetation models").

It is corrected as "*climate and vegetation models*".

Line 285: "Linearly interpolating"—Are there any proxy datasets available to constrain vegetation cover for the 2.5 ka period? How would different initial conditions affect the final simulated vegetation cover? Additionally, how can the derived paleo-vegetation be evaluated?

No, we don't have any proxy dataset to constrain the vegetation for 2.5ka.

As mentioned in the text, we interpolated linearly for 2.5ka from the mid-Holocene (6ka) vegetation proposed to reproduce the green Sahara conditions under PMIP4 protocol (Otto-Bliesner et al., 2017). The model simulation using this linearly interpolated vegetation, represented as shrubs and grass over North Africa and boreal forest over the northern high latitudes, substantially modified the surface albedo in the Northern Hemisphere. This vegetation–albedo feedback increased the pole-equator thermal gradient, shifting the ITCZ northward and consequently strengthening the African and South Asian monsoons (Singh et al., 2023).

The derived paleo vegetation represents the global climate response to the slowly varying orbital forcing which is, for time periods much shorter than the various cycles, effectively linear. Precessional cycles are 20 kyr. Obliquity cycles are 40 kyr. Eccentricity is 100 kyr. We have interpolated over 3 kyr.

Table 2: The reasoning behind the different simulation lengths across various experiments is unclear. What criteria were used to determine the duration of each experiment?

The varying simulation length across experiments primarily results from differences in the number of iterations performed for each setup. Based on the analysis, we used the following criteria to evaluate the model equilibrium during any iteration and convergence across the various iteration

1.) Model equilibrium: We evaluate the model equilibrium based on the condition that the absolute value of the decadal-mean planetary radiative imbalance should be < 0.2 W m$^{-2}$. We also evaluate the trend in global surface temperature, which (in absolute value terms) should be <0.1 °C/50 years within a 20-year moving window, and should change sign more than once across the last 100 years of equilibrium, which demonstrates oscillation around zero, rather than a small positive or negative bias.

2.) Convergence across each iteration: We evaluated model convergence across iterations by comparing the mean climate of the equilibrated model run with that of the previous iteration. For example, the plot below shows a comparison of the simulated annual surface temperature between iterations 4 and 3 of the experiment "2.5k_GS_BC_GISS". We performed similar evaluations for other diagnostics, such as ground albedo and net planetary radiation, as well as the spatial vegetation distributions, with each corresponding previous iteration.

[Figure]

Figure Mean surface temperature for iterations 4 (top) and 3 (middle) for the 2.5k_GS_BC_GISS simulation, and difference between the two iterations (bottom).

We added the following sentences in section 3.0:

*"Model equilibrium is determined using the threshold that the absolute value of the decadal-mean planetary radiative imbalance must be < 0.2 W m⁻², along with the surface temperature trend (absolute value <0.1 C/50 years). Convergence across iterations is evaluated by comparing the annual mean climate state and vegetation distributions between successive iterations."*

Figure 3: Is it reasonable to use present-day satellite-derived land cover for PI simulations? Please provide justification.

The tradeoff in doing paleoclimate and even early historical climate simulations is that the forcing changes to the climate system are relatively larger, but the constraining data sparser. Thus, unfortunately, observations of the entire world are somewhat lacking in the pre-satellite era and biased heavily towards Europe and North America. Thus, the background vegetation during the vegetation in pre-industrial times is prescribed globally by combining not only the global retrieval of vegetation, but also global estimates of fraction of anthropogenic land-use. The natural vegetation fractions remain the same per grid box, but anthropogenic (crop) land-use changes with time, which suppresses natural vegetation as crops expand (Matthews, 1983).

Lines 389-391: These sentences belong in the figure captions rather than the main text.

We believe that these lines are important, as they refer to the supplementary figures in context to the discussion of land cover changes under asynchronous coupling. No changes made.

Figure 10: The differences between LPJ and GISS, as well as between PI and GS, are relatively minor. The most significant differences appear between x and BC. This conclusion should be explicitly stated and explained in the main text.

Thanks for pointing this out. It suggests that internal variability is not substantially different between the two models (LPJ and GISS), thus their consequences are not noticeable during the asynchronous coupling. Whereas, in the case of partial greening over Sahara, the GISS model initially produced an increase in precipitation over North Africa (in the initial run), but this northward propagation of ITCZ over Africa didn't sustain in subsequent iterations or at the equilibrated simulation.

Thus, we added the following sentences in the abstract.

*"The asynchronously coupled model system shows strong vegetation-albedo feedback on climate and is comparatively more sensitive to the bias correction than the internal model variability and green Sahara conditions."*

We also added a sentence at the end of the abstract culminating the overall objective of this study. (L42).

*"This study presents a generalized framework for incorporating biogeophysical responses into climate models without dynamic vegetation, for simulating past climates, in line with the recommendations of the Paleoclimate Modelling Intercomparison Project (PMIP)"*

and similar statements highlighted this outcome in the section "6.0 Discussion and Conclusion" section (see L710 and L749)

Figure 11: In some regions, differences among simulations are substantial, while in others, they are minimal. What accounts for these regional variations? What role does vegetation play in shaping these differences?

The annual cycle of precipitation broadly shows an increase over the Indian subcontinent during the summer months and North American region during all peak precipitation months, and a decrease over the European regions during the summer months. The increase in precipitation over North America and India is more intense when bias correction is applied. This might be due to an increase in the equator-to-pole thermal gradient coupled with the vegetation-albedo feedback. In the case of partial greening over Sahara, the model initially produced an increase in precipitation over North Africa, but this northward propagation of ITCZ over Africa didn't sustain in subsequent iterations.

On the other hand, the intense decrease in summer precipitation over the European region is common among all experiments. This suggests the loss of needleleaf and cold deciduous broadleaf trees and increase of c3-grass might have contributed to the regional drying, instead of the increased warming the bias correction imposed. The loss of needleleaf trees is stronger in the absence of bias correction, resulting in a stronger decrease in European regional precipitation.

We added these sentences in the corresponding

*"Overall, the changes in annual precipitation cycle (increases or decreases) over the regions are primarily driven by both the pole-equator thermal gradients in the various experiments, as well as the biogeophysical effects associated with regional vegetation changes over these regions (e.g. Indian Summer monsoon, North American and European region) (Pausata et al., 2014; Tiwari et al., 2023; Singh et al., 2023)"*

Section 5: This section should be positioned before the analysis for better logical flow.

We moved Section 5 before the analysis; this is now named as section 4.

Figure 12: Proxy data appear to be more consistent with PI simulations than with the 2.5 ka simulation. What is the purpose of bias correction or climate-vegetation coupling if it does not improve the agreement with proxy data?

We apologize for the poor annotation in the submitted version of this figure, which caused the misconception that the PI simulations are better than the 2.5K ones. In Fig 12 (Figure 6 after revision), the 2.5k_PI_ctrl simulation which follows the PMIP4 protocol and uses present-day vegetation, it aligns right on top of the PI control line (1850_PI_ctrl; horizontal line in the submitted figure, and cyan-colored symbols in the modified figure below and the revised manuscript). That simulation does not compare as well with Kaufman's multi-proxy estimates for 2.5ka, whereas the experiments with the bias corrections (2.5k_PI_BC_LPJ and 2.5k_GS_BC_GISS; dark green and orange symbols) more closely match with Kaufman's estimates of mean difference from pre-industrial period for global (90S-90N) as well as over the tropical (0-30N and 0-30S) latitude bands.

We enhanced Figure 12 by adding the "1850_PI_ctrl" in the plot, and also modified the text under section 5.1 (section 4.1 after revisions) as given below.

[Figure]

Fig 12:[Figure 6 after revision] Comparison of model-simulated annual surface temperature anomalies and interannual variability for 2.5ka (with LPJ-LMfire vegetation) against the independent proxy-based temperature reconstructions (black, Kaufman et al., 2020). Mean (circle), median (line) along with 5-95 percentile range as variability bars (whiskers) and different colors represent the final iteration of our different experiments. (AFTER Revision it is figure 6 now)

*"It shows that the 2.5ka control simulation with present-day vegetation is comparable to pre-industrial conditions (1850_PI_ctrl), exhibiting a slightly cooler climate. In contrast, proxy-based surface temperature reconstructions (Kaufman et al., 2020) indicate slightly warmer conditions at global mean as well as across most latitude bands, except the far south (60S-90S). Applying bias correction allows the model to reproduce the same anomaly sign as the reconstruction, with minimal global (90N-90S) mean bias relative to the proxy data (2.5k_PI_BC_LPJ and 2.5k_GS_BC_GISS). Although the magnitude of warming remains higher at the northern hemisphere high latitudes, this framework demonstrates the improved capability of the model to reproduce reconstructions via incorporating biogeophysical effects of past vegetation by adopting a bias correction".*

Line 640: How is isotopic composition represented in the model simulations? Please provide details on its setup.

We added some citations for the implementation of isotopes in ModelE: "*The details of the implementation of water isotopes are described elsewhere (Schmidt 1998; Aleinov and Schmidt 2006; LeGrande et al 2006)*".  We also included the text below in section 5.2: (AFTER revision section 4.2)

*"The isotopic composition of oxygen in water, expressed as the ratio of $^{18}O$ to $^{16}O$ serves as a fundamental tracer for investigating changes in the hydrological cycle. This ratio is highly sensitive to regional climate conditions and to the processes that regulate the hydrological cycle, such as temperature, precipitation, and evaporation. ModelE2.1 includes a representation of the stable water isotopologues as passive tracers and the isotopic composition of precipitation can be diagnosed from the model output (Schmidt 1998; Aleinov and Schmidt 2006; LeGrande and Schmidt 2006)".*

**References (Both RC1 & RC2)**

Aleinov, I. and Schmidt, G. A.: Water isotopes in the GISS ModelE land surface scheme, Glob. Planet. Change, 51, 108–120, https://doi.org/10.1016/j.gloplacha.2005.12.010, 2006.

Charney, J., Quirk, W. J., Chow, S., and Kornfield, J.: A Comparative Study of the Effects of Albedo Change on Drought in Semi–Arid Regions, 1977.

Charney, J. G.: Dynamics of deserts and drought in the Sahel, Q. J. R. Meteorol. Soc., 101, 193–202, https://doi.org/10.1002/qj.49710142802, 1975.

Comas-Bru, L., Rehfeld, K., Roesch, C., Amirnezhad-Mozhdehi, S., Harrison, S. P., Atsawawaranunt, K., Ahmad, S. M., Brahim, Y. A., Baker, A., Bosomworth, M., Breitenbach, S. F. M., Burstyn, Y., Columbu, A., Deininger, M., Demény, A., Dixon, B., Fohlmeister, J., Hatvani, I. G., Hu, J., Kaushal, N., Kern, Z., Labuhn, I., Lechleitner, F. A., Lorrey, A., Martrat, B., Novello, V. F., Oster, J., Pérez-Mejías, C., Scholz, D., Scroxton, N., Sinha, N., Ward, B. M., Warken, S., Zhang, H., and SISAL Working Group members: SISALv2: a comprehensive speleothem isotope database with multiple age–depth models, Earth Syst. Sci. Data, 12, 2579–2606, https://doi.org/10.5194/essd-12-2579-2020, 2020.

DeFries, R. S., Field, C. B., Fung, I., Justice, C. O., Los, S., Matson, P. A., Matthews, E., Mooney, H. A., Potter, C. S., Prentice, K., Sellers, P. J., Townshend, J. R. G., Tucker, C. J., Ustin, S. L., and Vitousek, P. M.: Mapping the land surface for global atmosphere-biosphere models: Toward continuous distributions of vegetation's functional properties, J. Geophys. Res. Atmospheres, 100, 20867–20882, https://doi.org/10.1029/95JD01536, 1995.

Doughty, C. E., Loarie, S. R., and Field, C. B.: Theoretical Impact of Changing Albedo on Precipitation at the Southernmost Boundary of the ITCZ in South America, https://doi.org/10.1175/2012EI422.1, 2012.

Kaufman, D., McKay, N., Routson, C., Erb, M., Dätwyler, C., Sommer, P. S., Heiri, O., and Davis, B.: Holocene global mean surface temperature, a multi-method reconstruction approach, Sci. Data, 7, 201, https://doi.org/10.1038/s41597-020-0530-7, 2020.

Kim, Y., Moorcroft, P. R., Aleinov, I., Puma, M. J., and Kiang, N. Y.: Variability of phenology and fluxes of water and carbon with observed and simulated soil moisture in the Ent Terrestrial Biosphere Model (Ent TBM version 1.0.1.0.0), Biogeosciences, https://doi.org/10.5194/gmdd-8-5809-2015, 2015.

Köhler, P., Nehrbass-Ahles, C., Schmitt, J., Stocker, T. F., and Fischer, H.: A 156 kyr smoothed history of the atmospheric greenhouse gases $CO_2$, $CH_4$, and $N_2O$ and their radiative forcing, Earth Syst. Sci. Data, 9, 363–387, https://doi.org/10.5194/essd-9-363-2017, 2017.

LeGrande, A. N. and Schmidt, G. A.: Global gridded data set of the oxygen isotopic composition in seawater, Geophys. Res. Lett., 33, https://doi.org/10.1029/2006GL026011, 2006.

Loulergue, L., Schilt, A., Spahni, R., Masson-Delmotte, V., Blunier, T., Lemieux, B., Barnola, J.-M., Raynaud, D., Stocker, T. F., and Chappellaz, J.: Orbital and millennial-scale features of atmospheric CH4 over the past 800,000 years, Nature, 453, 383–386, https://doi.org/10.1038/nature06950, 2008.

Magi, B. I.: Global Lightning Parameterization from CMIP5 Climate Model Output, https://doi.org/10.1175/JTECH-D-13-00261.1, 2015.

Matthews, E.: Global Vegetation and Land Use: New High-Resolution Data Bases for Climate Studies, 1983.

Otto-Bliesner, B. L., Braconnot, P., Harrison, S. P., Lunt, D. J., Abe-Ouchi, A., Albani, S., Bartlein, P. J., Capron, E., Carlson, A. E., Dutton, A., Fischer, H., Goelzer, H., Govin, A., Haywood, A., Joos, F., LeGrande, A. N., Lipscomb, W. H., Lohmann, G., Mahowald, N., Nehrbass-Ahles, C., Pausata, F. S. R., Peterschmitt, J.-Y., Phipps, S. J., Renssen, H., and Zhang, Q.: The PMIP4 contribution to CMIP6 – Part 2: Two interglacials, scientific objective and experimental design for Holocene and Last Interglacial simulations, Geosci. Model Dev., 10, 3979–4003, https://doi.org/10.5194/gmd-10-3979-2017, 2017.

Pausata, F. S. R., Messori, G., and Zhang, Q.: Impacts of dust reduction on the northward expansion of the African monsoon during the Green Sahara period, Earth Planet. Sci. Lett., 434, 298–307, https://doi.org/10.1016/j.epsl.2015.11.049, 2016.

Pfeiffer, M., Spessa, A., and Kaplan, J. O.: A model for global biomass burning in preindustrial time: LPJ-LMfire (v1.0), Geosci. Model Dev., 6, 643–685, https://doi.org/10.5194/gmd-6-643-2013, 2013.

Schmidt, G. A.: Oxygen-18 variations in a global ocean model, Geophys. Res. Lett., 25, 1201–1204, https://doi.org/10.1029/98GL50866, 1998.

Schneider, R., Schmitt, J., Köhler, P., Joos, F., and Fischer, H.: A reconstruction of atmospheric carbon dioxide and its stable carbon isotopic composition from the penultimate glacial maximum to the last glacial inception, Clim. Past, 9, 2507–2523, https://doi.org/10.5194/cp-9-2507-2013, 2013.

Siegenthaler, U., Stocker, T. F., Monnin, E., Lüthi, D., Schwander, J., Stauffer, B., Raynaud, D., Barnola, J.-M., Fischer, H., Masson-Delmotte, V., and Jouzel, J.: Stable Carbon Cycle   Climate Relationship During the Late Pleistocene, Science, 310, 1313–1317, https://doi.org/10.1126/science.1120130, 2005.

Singh, R., Tsigaridis, K., LeGrande, A. N., Ludlow, F., and Manning, J. G.: Investigating hydroclimatic impacts of the 168–158 BCE volcanic quartet and their relevance to the Nile River basin and Egyptian history, Clim. Past, 19, 249–275, https://doi.org/10.5194/cp-19-249-2023, 2023.

Sitch, S., Smith, B., Prentice, I. C., Arneth, A., Bondeau, A., Cramer, W., Kaplan, J. O., Levis, S., Lucht, W., Sykes, M. T., Thonicke, K., and Venevsky, S.: Evaluation of ecosystem dynamics,

plant geography and terrestrial carbon cycling in the LPJ dynamic global vegetation model, Glob. Change Biol., 9, 161–185, https://doi.org/10.1046/j.1365-2486.2003.00569.x, 2003.

Stocker, B. D., Roth, R., Joos, F., Spahni, R., Steinacher, M., Zaehle, S., Bouwman, L., Xu-Ri, and Prentice, I. C.: Multiple greenhouse-gas feedbacks from the land biosphere under future climate change scenarios, Nat. Clim. Change, 3, 666–672, https://doi.org/10.1038/nclimate1864, 2013.

Thonicke, K., Spessa, A., Prentice, I. C., Harrison, S. P., Dong, L., and Carmona-Moreno, C.: The influence of vegetation, fire spread and fire behaviour on biomass burning and trace gas emissions: results from a process-based model, Biogeosciences, 7, 1991–2011, https://doi.org/10.5194/bg-7-1991-2010, 2010.

Tiwari, S., Ramos, R. D., Pausata, F. S. R., LeGrande, A. N., Griffiths, M. L., Beltrami, H., Wainer, I., de Vernal, A., Litchmore, D. T., Chandan, D., Peltier, W. R., and Tabor, C. R.: On the Remote Impacts of Mid-Holocene Saharan Vegetation on South American Hydroclimate: A Modeling Intercomparison, Geophys. Res. Lett., 50, e2022GL101974, https://doi.org/10.1029/2022GL101974, 2023.

---

## Author Comment (AC2)

We would like to thank both reviewers for their comments that helped improve our manuscript. Please see below replies to all comments. The reviewers' (RC2) comments are in black, our replies in blue, and any updated or new text from the revised manuscript in "*quoted blue italics*". All line numbers mentioned below are from the submitted manuscript.

**Reviewer 2.**

This manuscript introduces an asynchronous coupling framework between the NASA GISS ModelE atmosphere/climate model and the dynamic global vegetation model LPJ-LMfire. The framework is applied to mid- to late-Holocene (2.5 ka) simulations, exploring how dynamic vegetation and fire affect vegetation cover, albedo, and regional climate. The idea is timely and valuable: asynchronous coupling between atmosphere models and DGVMs can help advance palaeoclimate simulations and improve the realism of land–atmosphere interactions. However, it lacks sufficient novelty. In its current form, it lacks sufficient methodological transparency, reproducibility, and robustness in validation to meet GMD standards.

Major comments:

1. The coupling strategy must be documented in a way that allows others to reproduce it. The current description is too brief. Please provide a step-by-step description of the asynchronous loop (order of model runs, frequency of exchanges, number of iterations). A complete list of exchanged variables, with units, spatial grids, interpolation method, temporal aggregation, and post-processing. Table 1 can be expanded to contain these details, or a detailed flow chart can be added. Any bias correction, spin-up, or equilibration applied between iterations. How did you validate the model results fit with reality? All of these need to be shown and discussed.

The requested details are already included in the manuscript in detail, following the logical flow of the modeling framework described in Section 2 (*Models and Methodology*) and Section 3 (*Experiment Designs*). This is clarified below, including some additional details we included in the revised manuscript.

Please provide a step-by-step description of the asynchronous loop (order of model runs, frequency of exchanges, number of iterations). These details are already included in Section 2.3: the flow of the asynchronous coupling is visualized in Fig. 1; a stepwise description is provided in lines 185-267; the post-processing details are mentioned along with each step in the framework description. However, added more text to clarify the specific points made by the reviewer.

order of model runs: [in section 3 at L282 and L287] "*Run '1850_PI_ctrl' (row 1 in table 2) was performed to evaluate the vegetation mapping scheme and to select the appropriate scheme for asynchronous coupling, whereas '2.5k_PI_ctrl' (row 2 in table 2) is used as the $0^{th}$ order control run for 2.5ka period with present-day vegetation distribution*"

"*....and extended the $0^{th}$ order 2.k control ('2.5k_PI_ctrl') before branching out the experiments '2.5k_GS_x_GISS' and '2.5k_GS_BC_GISS'*"

Frequency of exchange: Figure 1[Flow diagram] is modified to illustrate this detail.

number of iterations : This already exists in Table 2 column 5.

Units of exchange variables: Units are added to the columns 1 and 2 of Table 1.

Spatial grid and interpolation method: Already listed in Figure 1 (flow diagram) and L 264-266, L218, L285.

Postprocessing : Postprocessing details for both sets of data are already described in respective steps of the asynchronous coupling between GISS-ModelE and LPJ (see sections 2.3.1, 2.3.2, 2.3.3 and 2.3.4).

Table 1 can be expanded to contain these details, or a detailed flow chart can be added. Any bias correction, spin-up, or equilibration applied between iterations. A flow chart (Figure 1) is provided under the framework description at L190. A single table for all these had too much information due to the many different steps, thus we decided to separate these into 2 in the submitted manuscript, based on a) directly related with the asynchronous coupling framework and b) experimental setup chosen.

How did you validate the model results fit with reality? Section 3.1 (Evaluation and Validation of mapping methodologies) already covers the model validation. Also, in section 5 of the manuscript, we performed an extensive model-data comparison.

Other relevant finer details related with the PFTs mapping schemes and the initial vegetation distribution is already provided in the supplement. However, motivated by the reviewer's comments above, we modified the paragraph at the end of the introduction by summarizing the relevant details at L114, as given below. Further, we modified some section titles too: L120: Modified section 2 title as "Models and asynchronous coupling framework"; L162: modified section 2.2 title as "2.5ka simulation setup (initial control run using ModelE)".

*"Section 2 describes the models used in this study (Section 2.1), the initial control run for 2.5ka, and a stepwise description of the asynchronous coupling framework, including variable exchange and processing (Sections 2.2 and 2.3). Section 3 presents the experimental design for implementing the asynchronous coupled system and evaluates the PFT mapping schemes. In Section 4, we evaluate the simulated 2.5ka climate using the ModelE–LPJ asynchronous coupling framework against multi-proxy temperature reconstructions (Kaufman et al., 2020) and additionally utilized the model's capabilities to simulate the isotopic composition of water in precipitation ($\delta^{18}Op$) to compare with the Speleothem Isotope Synthesis and Analysis (SISAL) version 2 database (Comas-Bru et al., 2020). Section 5 provides the analysis and comparison of model-simulated climate under various experimental configurations."*

2. The manuscript does not clearly specify how many experiments/ensembles were conducted, nor their lengths. Please add a concise experiment table with experiment names, forcings applied, spin-up duration, ensemble size, and control/baseline. Clarify whether ensemble spread reflects internal variability or parameter uncertainty.

All the requested details are explicitly provided in Section 3 (line 269). Specifically, information regarding the experimental design, including their duration, number of iterations, applied forcings, and constraints etc. are already presented in Table 2.
The simulated spread of our results reflects the modeled interannual (internal) variability, not the model's parametric uncertainty. We now mention that in the text: L531 and several other places in section 4.1 *"5-95 percentile range (interannual variability)"*.

3. State explicitly which CO2/greenhouse gas values and orbital forcings were used for each experiment, with citations to PMIP4 or other sources. A dedicated table listing GHG concentrations would be helpful.

This experiment uses time period–specific forcings (2.5 ka and PI), including GHG concentrations ($CO_2$, $N_2O$, $CH_4$) and orbital configuration, which are explicitly described in Section 2.2 (lines 167-171). Section 2.2 was specifically designed to consolidate all relevant time-specific details, and since these are just a few numbers, We also added the forcing details in table 2 (row 1 & 2) for better comparison.

4. The choice of monthly coupling is not justified. Either show sensitivity to coupling interval (e.g., monthly vs seasonal/annual) or provide a discussion, with references, of the limitations and expected impacts of this choice.

The coupling between ModelE and LPJ-LMfire is based on monthly climatological means, i.e., multi-year monthly averages. ModelE uses static vegetation distribution fields from LPJ-LMfire as boundary conditions for the subsequent runs, which remain constant across monthly and interannual timescales. The models are designed to update their vegetation boundary conditions on a monthly basis, consistent with the standard approach used in CMIP and paleoclimate simulations (Otto-Bliesner et al., 2017). Altering vegetation distribution seasonally or even annually would not capture vegetation variability with sufficient temporal resolution. Moreover, there is not enough information available to implement changes more frequently than monthly.

We modified figure 1 (flow diagram) to specify the temporal frequencies of inputs to LPJ and GISS ModelE.

5. The manuscript compares simulations with reconstructions but lacks detail on datasets and metrics. Please list all proxy datasets used, with version numbers and coverage. Describe the model-data comparison method (regridding, seasonality, temporal windows). Provide quantitative metrics (bias, RMSE, correlation, significance tests) and report proxy uncertainties.

The submitted manuscript compares the model simulated 2.5ka climate against the global mean surface temperature (GMST) and a latitude band of 30 degree from 90S to 90N from Kaufman et al. (2020) as shown in figure 12 (now fig 6 after revision). All the necessary data and related information such as total number of proxies (L608) from Kaufman et al. (2020) that have been used, including how we estimated the different uncertainty metrics using the estimates from 5 methods used in Kaufman et al. (2020), are already explained in L614-L617. More details can be found in Kaufman et al. (2020), and we believe it does not add value to repeat them in this manuscript.
Similarly, the details of Speleothem Isotope Synthesis and Analysis (SISAL) version 2 database (Comas-Bru et al., 2020), including the number of sites for the 2.5ka period, is provided in section 5.2 (now section 4.2 after revision). Specifically, lines 641-649 describe the methodology for the site-based comparison with model simulations, while details of statistics (RMSE, correlation, standard deviation) are thoroughly discussed across section 5.2. Figure 13 (Section 4.2 after revisions), Figure 14 and Figure 15 clearly support the summarized discussion related to site locations, regional demarcation and the statistical estimates for the regional comparison of model simulated climate conditions for 2.5ka period. Figure S4 also adds to the discussion here.

Regridding, seasonality and temporal window: We assume that this question is regarding the model output and reconstruction comparison. These specifics were not directly related with the proxy reconstructions we used here, however for the Kaufman et al. (2020) temperature reconstruction is only available as the latitudinal band and global means. Regarding the SISAL v2, which is a site-based data and the comparison methodology is described in detail in section 5.2 (After revision 4.2).

6. Discuss uncertainties more systematically: (i) forcings, (ii) internal variability, and (iii) mapping of LPJ PFTs to ModelE classes. If multiple mapping schemes were tested, please include the results or add them to the Supplement.

(i) forcings: We used the time periods (2.5k and preindustrial) specific GHG, Ozone and orbital forcings from the literature and this aspect is discussed there (Köhler et al., 2017; Loulergue et al., 2008; Otto-Bliesner et al., 2017; Schneider et al., 2013; Siegenthaler et al., 2005). Evaluating the impacts of the uncertainty in these forcings is beyond the scope of this study.

(ii) internal variability : To account for internal variability, we performed a Student t-test to assess statistical significance at the 5-95% confidence level with respect to the 100-year equilibrated control run. Stippling on the spatial plots indicates regions that are not statistically significant. Thus we only discussed the statistically significant changes.

(iii) mapping: Section 3.1 describes how we used different LPJ-LMfire-to-ModelE PFT mapping schemes and evaluated them for the preindustrial period. A detailed discussion on how various mapping schemes influence the surface temperature, rainfall and albedo is also presented. Further, the finalized mapping scheme is compared against the land cover used for preindustrial simulations. A supplementary table TS1 is included to summarize the details of different mapping schemes.

7. Clarify how fire is represented in LPJ-LMfire for these experiments: which ignition sources are included, how burned area is calibrated, and whether fire emissions feedback to ModelE. If emissions are excluded, state this explicitly.

Only lightning ignitions were included in the LPJ-LMfire simulations. Emissions of greenhouse gases and aerosols from simulated wildfires do not give feedback on the ModelE simulations as this would require the representation of atmospheric chemistry and transport processes that are not included in the version of ModelE we used.
We added the following in section 2.3.2:

*"Lightning density was estimated based on modelled convective mass flux following Magi (2015). However, the feedback to climate due to fire-driven emissions are not included, as accounting for them would require active atmospheric chemistry and transport, which are not included in LPJ-LMfire".*

8. Improve figure captions so each is interpretable without referring to the main text. All figures must include units, baseline (difference relative to what), averaging period, and statistical significance.

We revisited all figure captions to include the relevant details.

Minor comments:

1. Be consistent with "2.5 ka" wording – clarify whether you mean 2.5 ka BP (2500 years before present), 2.5kyr, or "mid-Holocene" and give exact calendar years used.

Thanks for pointing this out. We modified the text to be consistent with 2.5 ka and mid-Holocene (6 ka).

2. The manuscript lists a few CO2 values (e.g., 271.4 ppm, ~279 ppm, 284 ppm). Make sure each value is linked to the correct experiment and source citation. Possibly a single table listing GHGs for each experiment would be useful.

Thanks for pointing this out, we modified the citation (Köhler et al., 2017) and since these are few values for specific time periods, We also added the GHG forcings details into table 2 (row 1 and 2).

3. You mention testing several LPJ-GISS PFT mapping schemes. Please show results (or at least diagnostics) illustrating how sensitive your key climate metrics are to the mapping choice.

Section 3.1 presents the analysis for the 3 mapping schemes we compared, and details about them are presented in table TS1 of the submitted manuscript.

4. A few citations seem incomplete or inconsistently formatted. Please check the references for completeness and adhere to the journal style.

Thanks for pointing this out, we made all necessary corrections.

5. In the discussion, the text implies strong climate impacts from the coupling that are not fully supported by the presented evidence. Tone down causal language where results are suggestive rather than robust and clearly label speculative statements.

In the discussion section, at L710 "strong influence" refers to the influence of including/excluding the bias correction in the coupling process. This is justified as the model drift towards colder climate conditions in the absence of a bias correction. However, we replaced this with the more suitable word "pronounced".

**References (Both RC1 & RC2)**

Aleinov, I. and Schmidt, G. A.: Water isotopes in the GISS ModelE land surface scheme, Glob. Planet. Change, 51, 108–120, https://doi.org/10.1016/j.gloplacha.2005.12.010, 2006.

Charney, J., Quirk, W. J., Chow, S., and Kornfield, J.: A Comparative Study of the Effects of Albedo Change on Drought in Semi–Arid Regions, 1977.

Charney, J. G.: Dynamics of deserts and drought in the Sahel, Q. J. R. Meteorol. Soc., 101, 193–202, https://doi.org/10.1002/qj.49710142802, 1975.

Comas-Bru, L., Rehfeld, K., Roesch, C., Amirnezhad-Mozhdehi, S., Harrison, S. P., Atsawawaranunt, K., Ahmad, S. M., Brahim, Y. A., Baker, A., Bosomworth, M., Breitenbach, S. F. M., Burstyn, Y., Columbu, A., Deininger, M., Demény, A., Dixon, B., Fohlmeister, J., Hatvani, I. G., Hu, J., Kaushal, N., Kern, Z., Labuhn, I., Lechleitner, F. A., Lorrey, A., Martrat, B., Novello, V. F., Oster, J., Pérez-Mejías, C., Scholz, D., Scroxton, N., Sinha, N., Ward, B. M., Warken, S., Zhang, H., and SISAL Working Group members: SISALv2: a comprehensive speleothem isotope database with multiple age–depth models, Earth Syst. Sci. Data, 12, 2579–2606, https://doi.org/10.5194/essd-12-2579-2020, 2020.

DeFries, R. S., Field, C. B., Fung, I., Justice, C. O., Los, S., Matson, P. A., Matthews, E., Mooney, H. A., Potter, C. S., Prentice, K., Sellers, P. J., Townshend, J. R. G., Tucker, C. J., Ustin, S. L., and Vitousek, P. M.: Mapping the land surface for global atmosphere-biosphere models: Toward continuous distributions of vegetation's functional properties, J. Geophys. Res. Atmospheres, 100, 20867–20882, https://doi.org/10.1029/95JD01536, 1995.

Doughty, C. E., Loarie, S. R., and Field, C. B.: Theoretical Impact of Changing Albedo on Precipitation at the Southernmost Boundary of the ITCZ in South America, https://doi.org/10.1175/2012EI422.1, 2012.

Kaufman, D., McKay, N., Routson, C., Erb, M., Dätwyler, C., Sommer, P. S., Heiri, O., and Davis, B.: Holocene global mean surface temperature, a multi-method reconstruction approach, Sci. Data, 7, 201, https://doi.org/10.1038/s41597-020-0530-7, 2020.

Kim, Y., Moorcroft, P. R., Aleinov, I., Puma, M. J., and Kiang, N. Y.: Variability of phenology and fluxes of water and carbon with observed and simulated soil moisture in the Ent Terrestrial Biosphere Model (Ent TBM version 1.0.1.0.0), Biogeosciences, https://doi.org/10.5194/gmdd-8-5809-2015, 2015.

Köhler, P., Nehrbass-Ahles, C., Schmitt, J., Stocker, T. F., and Fischer, H.: A 156 kyr smoothed history of the atmospheric greenhouse gases $CO_2$, $CH_4$, and $N_2O$ and their radiative forcing, Earth Syst. Sci. Data, 9, 363–387, https://doi.org/10.5194/essd-9-363-2017, 2017.

LeGrande, A. N. and Schmidt, G. A.: Global gridded data set of the oxygen isotopic composition in seawater, Geophys. Res. Lett., 33, https://doi.org/10.1029/2006GL026011, 2006.

Loulergue, L., Schilt, A., Spahni, R., Masson-Delmotte, V., Blunier, T., Lemieux, B., Barnola, J.-M., Raynaud, D., Stocker, T. F., and Chappellaz, J.: Orbital and millennial-scale features of

atmospheric CH4 over the past 800,000 years, Nature, 453, 383–386, https://doi.org/10.1038/nature06950, 2008.

Magi, B. I.: Global Lightning Parameterization from CMIP5 Climate Model Output, https://doi.org/10.1175/JTECH-D-13-00261.1, 2015.

Matthews, E.: Global Vegetation and Land Use: New High-Resolution Data Bases for Climate Studies, 1983.

Otto-Bliesner, B. L., Braconnot, P., Harrison, S. P., Lunt, D. J., Abe-Ouchi, A., Albani, S., Bartlein, P. J., Capron, E., Carlson, A. E., Dutton, A., Fischer, H., Goelzer, H., Govin, A., Haywood, A., Joos, F., LeGrande, A. N., Lipscomb, W. H., Lohmann, G., Mahowald, N., Nehrbass-Ahles, C., Pausata, F. S. R., Peterschmitt, J.-Y., Phipps, S. J., Renssen, H., and Zhang, Q.: The PMIP4 contribution to CMIP6 – Part 2: Two interglacials, scientific objective and experimental design for Holocene and Last Interglacial simulations, Geosci. Model Dev., 10, 3979–4003, https://doi.org/10.5194/gmd-10-3979-2017, 2017.

Pausata, F. S. R., Messori, G., and Zhang, Q.: Impacts of dust reduction on the northward expansion of the African monsoon during the Green Sahara period, Earth Planet. Sci. Lett., 434, 298–307, https://doi.org/10.1016/j.epsl.2015.11.049, 2016.

Pfeiffer, M., Spessa, A., and Kaplan, J. O.: A model for global biomass burning in preindustrial time: LPJ-LMfire (v1.0), Geosci. Model Dev., 6, 643–685, https://doi.org/10.5194/gmd-6-643-2013, 2013.

Schmidt, G. A.: Oxygen-18 variations in a global ocean model, Geophys. Res. Lett., 25, 1201–1204, https://doi.org/10.1029/98GL50866, 1998.

Schneider, R., Schmitt, J., Köhler, P., Joos, F., and Fischer, H.: A reconstruction of atmospheric carbon dioxide and its stable carbon isotopic composition from the penultimate glacial maximum to the last glacial inception, Clim. Past, 9, 2507–2523, https://doi.org/10.5194/cp-9-2507-2013, 2013.

Siegenthaler, U., Stocker, T. F., Monnin, E., Lüthi, D., Schwander, J., Stauffer, B., Raynaud, D., Barnola, J.-M., Fischer, H., Masson-Delmotte, V., and Jouzel, J.: Stable Carbon Cycle   Climate Relationship During the Late Pleistocene, Science, 310, 1313–1317, https://doi.org/10.1126/science.1120130, 2005.

Singh, R., Tsigaridis, K., LeGrande, A. N., Ludlow, F., and Manning, J. G.: Investigating hydroclimatic impacts of the 168–158 BCE volcanic quartet and their relevance to the Nile River basin and Egyptian history, Clim. Past, 19, 249–275, https://doi.org/10.5194/cp-19-249-2023, 2023.

Sitch, S., Smith, B., Prentice, I. C., Arneth, A., Bondeau, A., Cramer, W., Kaplan, J. O., Levis, S., Lucht, W., Sykes, M. T., Thonicke, K., and Venevsky, S.: Evaluation of ecosystem dynamics, plant geography and terrestrial carbon cycling in the LPJ dynamic global vegetation model, Glob. Change Biol., 9, 161–185, https://doi.org/10.1046/j.1365-2486.2003.00569.x, 2003.

Stocker, B. D., Roth, R., Joos, F., Spahni, R., Steinacher, M., Zaehle, S., Bouwman, L., Xu-Ri, and Prentice, I. C.: Multiple greenhouse-gas feedbacks from the land biosphere under future climate change scenarios, Nat. Clim. Change, 3, 666–672, https://doi.org/10.1038/nclimate1864, 2013.

Thonicke, K., Spessa, A., Prentice, I. C., Harrison, S. P., Dong, L., and Carmona-Moreno, C.: The influence of vegetation, fire spread and fire behaviour on biomass burning and trace gas emissions: results from a process-based model, Biogeosciences, 7, 1991–2011, https://doi.org/10.5194/bg-7-1991-2010, 2010.

Tiwari, S., Ramos, R. D., Pausata, F. S. R., LeGrande, A. N., Griffiths, M. L., Beltrami, H., Wainer, I., de Vernal, A., Litchmore, D. T., Chandan, D., Peltier, W. R., and Tabor, C. R.: On the Remote Impacts of Mid-Holocene Saharan Vegetation on South American Hydroclimate: A Modeling Intercomparison, Geophys. Res. Lett., 50, e2022GL101974, https://doi.org/10.1029/2022GL101974, 2023.